# Impact of vertical wind shear on roll structure in idealized hurricane boundary layers

Shouping Wang and Qingfang Jiang

Naval Research Laboratory, Monterey, CA 93943, USA

*Correspondence to*: Shouping Wang (shouping.wang@nrlmry.navy.mil)

**Abstract.** Quasi two-dimensional roll vortices are frequently observed in hurricane boundary layers. It is believed that this highly coherent structure, likely caused by the inflection point instability, plays an important role in organizing turbulent transport. Large-eddy simulations are conducted to investigate the impact of wind shear characteristics such as the shear strength and inflection-point level on the roll structure in terms of its spectral characteristics and turbulence organization. A mean wind nudging approach is used in the simulations to maintain the specified mean wind shear without directly affecting turbulent motions. Enhancing the radial wind shear expands the roll horizontal scale and strengthens the roll's kinetic energy. Increasing the inflection-point level tends to produce a narrow and sharp peak in the power spectrum at the wavelength consistent with the roll spacing indicated by the instantaneous turbulent fields. The spectral tangential momentum flux, in particular, reaches a strong peak value at the roll wavelength. In contrast, the spectral radial momentum flux obtains its maximum at the wavelength that is usually shorter than the roll's, suggesting that the roll radial momentum transport is less efficient than the tangential because of the quasi two dimensionality of the roll structure. The most robust rolls are produced in a simulation with the highest inflection-point level and relatively strong radial wind shear. Based on the spectral analysis, the roll-scale contribution to the turbulent momentum flux can reach 40 % in the middle of the boundary layer.

## 1 Introduction

The hurricane boundary layer (HBL) is well known for its critical role in evolutions of tropical cyclones (TCs) as the air-sea interaction represents both the most important source and sink of the moist available energy and the kinetic energy, respectively. One of the frequently occurring features in the HBL is horizontal roll vortices, which have quasi-two dimensional coherent and banded structure extending from the surface to the top of the HBL. The observed horizontal roll scale, i.e., the average distance between two neighbouring rolls, ranges from sub-kilometre to ~ 10 km (Wurman and Winslow, 1998; Lorsolo et al., 2008; Foster, 2013). Observational and modelling studies suggest that these roll vortices make significant contribution to the vertical heat and momentum transport (Zhang et al., 2008; Zhu, 2008) and thus provide a critical control of the wind, temperature and moisture profiles.

Previous studies have attributed the prevalence of the roll structure to the existence of an inflection point in the mean HBL radial wind profile and attempted to establish the link between the HBL environment and the roll statistical characteristics (e.g. Foster, 2005; Nolan, 2005). These analyses are generally consistent with observations: 1) the rolls are oriented at 0-10° to the left of the tangential wind; 2) the roll aspect ratio (ratio of the horizontal scale to the vertical) ranges from 2 to 4; and 3) the roll generated momentum fluxes are nonlocal. A recent study by Foster (2013) differentiates the standard boundary layer roll vortices, as those highlighted above, from the observed large roll vortices from Synthetic Aperture Radar images, whose horizontal scale reaches

to 10-20 km. His results from a two-dimensional nonlinear resonant triad interaction model further suggest that the observed unusually large roll aspect ratio result from the up-scale energy transport through the nonlinear wave-wave interaction. Gao and Ginis (2014, hereafter GG14; 2016) investigated the formation of HBL rolls by solving a two-dimensional perturbation system driven by the mean wind profiles that are the solutions of an axisymmetric HBL model. They concluded that the mean wind shear intensity affects the roll growth rate and the inflection-point level (IPL hereafter) impacts the roll wavelength. While these two-dimensional quasi-analytical models have significantly advanced our understanding of HBL roll dynamics, they cannot accurately represent three dimensional stochastic turbulent flows. These work and conclusions are worth revisiting using a large-eddy simulation (LES) approach.

There have been a few LES studies of HBL rolls. Zhu (2008) configured a nested WRF (Weather Research Forecast) model to include an LES domain with a horizontal resolution of 100 m and a vertical grid spacing varying from 5 to 65 m below 1.6 km. The WRF-LES was used to simulate a real case of hurricane landfall. Organized large-eddy circulations with horizontal scales ranging from 1 to 10 km were found to intensely enhance the vertical momentum, heat and moisture transport. He further proposed a framework of the turbulent transport parameterization based on the conceptual model of convective up- and down-draft representation for shallow cumulus convection. While this mesoscale-LES grid nesting framework represents a realistic and sophisticated numerical approach, it does not allow for sensitivity studies to examine impact of various mean conditions, such as wind profiles, on the roll structure. In an idealized study of HBL rolls, Nakanishi and Niino (2012, hereafter NN12) adopted a traditional LES approach, which uses a $20 \times 20 \times 4$ km$^3$ domain with periodic lateral boundary conditions. They concluded that the inflection point instability in the radial wind profile leads to the formation of the quasi-linear roll structure with wavelengths between 1.5 and 2.4 km. The LES study by Green and Zhang (2015) also confirmed many of these findings and further suggested that the turbulence diffusivity varies considerably among different simulations, an indication that the downgradient transfer model breaks down for the momentum fluxes associated with HBL rolls.

Among these LES studies, only the WRF-LES nesting approach used by Zhu (2008) explicitly simulates mesoscale circulations and thus their effects on the roll structure. Others neglect the horizontal advection effects by assuming a local balance among the turbulent mixing, gradient wind, Coriolis force, and hurricane induced centripetal force. Consequently, the wind profile based on the local force balance may not represent the most relevant features with respect to the roll development in the HBL in the LES studies. For example, Morrison et al. (2005) provided both observed radial and tangential winds from WSR-88D radar data, and the inflection point levels (IPLs) estimated from these observations are about 300 m to 800 m for the winds at the tropical cyclone (TC) radius 29 km to 122 km, respectively. These IPLs are generally higher than those of the LES simulations by NN12 which are 100 m and 300 m at the radius 40 and 100 km, respectively. Therefore, there is a need to use more realistic wind profiles in the LES studies. The latest study of Bryan et al (2017) provided an improved HBL LES framework that accounts for the influence of mesoscale advections on the wind profiles. The current work introduces an empirical approach as discussed in the next section.

Boundary layer rolls have been a subject of many studies since 1960s as reviewed by Atkinson and Zhang (1996) and Young et al (2002). Several physical mechanisms have been proposed for different environments, including combined surface shear-buoyancy instability (Moeng and Sullivan, 1994, Glendening, 1996), the surface shear-cloud convection-radiation instability (Chlond, 1992), parallel instability (Lilly, 1966), and inflection point instability (Brown, 1970; Brown, 1972; Foster, 2005). As discussed at the beginning of the paper, the most relevant mechanism for the HBL rolls is the inflection point instability. This

work aims to gain new understanding of the impact of the mean wind profile characteristics that are directly associated with the inflection point instability, the radial wind shear and IPL, on the roll structure. We use a different LES approach, featuring a mean nudging method which is applied to the momentum equations to strongly regulate the mean wind profile. This approach enables us to conduct a systematic study of the roll response, including the growth of the HBL, turbulence intensity, and the spectral distribution, to changes in the mean wind profiles. The remainder of the paper is organized as follows. Section 2 describes the LES model and simulation setup. Sections 3 and 4 provide general description of the simulation results and spectral analysis, respectively. Further discussions on the wind shear are given in section 5. Section 6 summarizes the work.

## 2 Approach

### 2.1 COAMPS-LES

The Naval Research Laboratory Coupled Ocean/Atmosphere Mesoscale Prediction System-Large-Eddy Simulation (COAMPS-LES) is used in this study. The LES model was first introduced by Golaz et al. (2005) for the study of boundary layer cloud systems. It has been applied to investigate various types of boundary layer turbulence, including topographic flows, and stratocumulus dynamics (Golaz et al., 2009; Wang et al., 2012; Jiang and Wang, 2013). Readers are referred to these papers for detailed description as well as its various applications. Briefly, the model applies the anelastic approximation for efficient numerical computation and uses the Deardorff's prognostic turbulence kinetic energy approach for the subgrid-scale model (Deardorff, 1980). The model coordinate is configured such that $x$ is directed away from the centre of a TC in the radial direction, $y$ is in the direction 90° counter-clockwise from $x$, i.e., the azimuthal or tangential direction, and $z$ the vertical axis. Because our simulations are focused on the dynamics and structure of the rolls, moisture is not included. The predictive variables are radial wind $u$, tangential wind $v$, potential temperature $\theta$, and subgrid-scale turbulence kinetic energy. The model uses the horizontal resolution $\Delta x = \Delta y = 50$ m and a variable vertical grid with $\Delta z = 30$ m below 3km gradually increasing to 200 m. This grid covers a $25.2 \times 25.2 \times 4.9$ km$^3$ domain. The Rayleigh damping technique is applied near the model top to reduce downward reflection of internal gravity waves. The surface momentum flux is calculated using the roughness length ($z_0$) formulation of Donelan et al. (2004). That is, $z_0$ increases with the 10 m wind speed following the Charnock relationship for the wind speed less than 33 m s$^{-1}$, above which $z_0$ is set equal to 3.35 mm, which is equivalent to a drag coefficient of 0.0025. Because the 10 m wind is usually less than 33 m s$^{-1}$ for all the simulations, this modification of $z_0$ on the Charnock relationship should not have major effects on the results presented here. To accelerate the LES spin-up process, a moderate constant surface heat flux $F_h = 20$ W m$^{-2}$ is applied. Because of the strong near-surface winds ($\sim 30$ m s$^{-1}$), the application of the heat flux does not change the dominance of the shear production of turbulence. For comparison purposes, all the simulations start with the same initial conditions. The horizontal wind is specified as a constant gradient wind speed ($V_g$) and the linear potential temperature ($\theta$) profile with a gradient of 0.00475 K m$^{-1}$ and a value of 298.5 K at the first vertical level. The gradient wind $V_g$ is fixed at 45.5 m s$^{-1}$ as the value represents a middle-to-high speed range in a hurricane environment (e.g. Willoughby, 1990). The model is integrated for 10 h with a time step of 0.5 s.

### 2.2 Mean wind nudging

As discussed in the introduction, the mean wind profiles from the LES simulations that do not include the mesoscale circulations (e.g. HBL inflow) may not adequately represent the wind characteristics in a hurricane environment. It is highly desirable that observationally based wind profiles be used and approximately maintained throughout the simulations. We adopt a modelling approach that strongly regulates the mean wind profile according to our specifications. A special relaxation term is added to

each horizontal momentum equation to nudge the mean wind toward a specified target wind profile. A unique feature of these nudging terms is that they only nudge the horizontally averaged wind. That is, at each time step, the horizontal mean wind profile, which is dependent only on $z$, is calculated from the predicted winds and used as the variable in the nudging term. Because the target profile is only a function of $z$, the nudging tendency is exactly the same at every horizontal grid point for the same level at each time step. Consequently, the LES simulated turbulent perturbations, which are defined as deviates from a horizontal mean, are not directly affected by the nudging terms. Both the turbulent perturbations and statistics are, of course, regulated by the mean wind profiles. This nudging approach was used to spin up LES simulations of stratocumulus clouds by Kazil et al. (2016).

The momentum equations with the nudging terms can be written as

$$\frac{\partial u}{\partial t} = -\vec{v}\cdot\nabla u - \frac{1}{\rho_0}\frac{\partial p}{\partial x} + SGS - \left[\frac{V_g^2 - v^2}{R} - f(V_g - v)\right] + \left\{\frac{U_T(z) - \langle u\rangle(z)}{\tau}\right\} \tag{1}$$

and

$$\frac{\partial v}{\partial t} = -\vec{v}\cdot\nabla v - \frac{1}{\rho_0}\frac{\partial p}{\partial y} + SGS - \left[\frac{u\cdot v}{R} + fu\right] + \left\{\frac{V_T(z) - \langle v\rangle(z)}{\tau}\right\}, \tag{2}$$

where $R$ is the radius from the LES domain to the centre of the TC, $U_T$ and $V_T$ are the prescribed target radial and tangential wind components, respectively, $\rho_0$ is the air density of an atmospheric reference state, $V_g$ denotes the gradient wind, $SGS$ represents effects of subgrid scale motions, $\langle \rangle$ is a horizontally averaged variable at each time step, and $\tau$ is a relaxation time scale. Other symbols in Eq. (1) and (2) have their generally accepted meaning. These equations are the same as those used by NN12 except the relaxation terms represented by the curly bracket in each equation. The square bracket is the gradient wind imbalance term associated with the centripetal force, the Coriolis force, and the large-scale radial pressure gradient. This term represents a major forcing that is responsible for the mean wind shear characteristics; it is designated as the rotational term hereafter for simplicity. Sensitivity simulations have been conducted to evaluate how well the mean wind profiles can be controlled by the nudging term. We find that the mean wind profiles are better regulated by the nudging if the rotational terms are removed and its removal has little impact on the turbulence statistics. This is consistent with the previous studies showing negligible effects from the curvature terms on the roll structure as well as the turbulence generation in general (Foster, 2005; NN12). Thus the square bracket terms are set zero in Eq. (1) and Eq. (2) in this paper unless specified otherwise. Therefore, the nudging terms are used to represent all the major processes that control the mean wind profiles, except turbulence mixing. It is noteworthy that this new approach has a number of attractive advantages. Firstly, it maintains the mean wind profiles, which are derived from observations or balanced dynamic models, and accordingly, are more realistic. Secondly, it offers a convenient way to systematically change the mean wind profile and therefore, allows us to examine the roll's response to these changes. At last, because the actual rotational terms are not explicitly included in the momentum equations, the horizontal winds no longer rotate with time and LES simulations may reach non-oscillatory quasi-equilibrium solutions. A comparison of three test simulations is presented in Appendix.

### 2.3 Target wind profiles

We are interested in two sets of LES quasi-equilibrium solutions corresponding to different mean wind characteristics with regard to both the wind shear strength and IPL. These two parameters are chosen because according to previous studies, they are

key parameters related to inflection-point instability. The former is the main source of turbulence and the latter is linked to the roll scales (e.g. Chlond, 1992; GG14). The vertical shear of the radial wind above the surface layer is a main focus of this study. The shear layer, where the inflection point is located, usually extends from ~ 100 m to the top of the HBL. To avoid confusion, we use the term "surface wind shear" to describe the wind shear that is concentrated in the lowest 100 m.

The target wind profiles are formulated based on the normalized typical hurricane wind profiles obtained from a dynamical model of Foster (2005) and from the observations by Morrison et al. (2005). The LES mean winds are nudged toward the target profiles, which are formulated to represent various wind shear conditions. This approach facilitates the study of the response of roll formation and dynamics to wind profiles through sensitivity simulations. We have experimented with dozens of LES simulations using a variety of target wind profiles. The two groups of the target wind profiles (i.e., groups H and L, see Fig. 1) are chosen from these additional trial simulations and they exhibit systematic variations in shear strength and infection point levels. The target radial wind $U_T$ of H2 and tangential wind $V_T$ of group H generally follow those of Fig. 2 of Foster (2005) except for the HBL height. In addition, the super gradient wind shape is also included in $V_T$ in accordance to Fig 3a of Morrison et al. (2005). The $U_T$ profile of H2 is multiplied by 0.5 and 1.5 to provide $U_T$ for H1 and H3, respectively. The target radial wind $U_T$ of L2 is obtained by vertically suppressing $U_T$ of H2 and increasing the near-surface value to 13 m s$^{-1}$. Then, $U_T$ of L2 is multiplied by 0.5 and 1.5 to give $U_T$ of L1 and L3, respectively. The target tangential wind profile $V_T$ of group L is obtained by lowering the HBL height for $V_T$ of group H.

In summary, group L simulations are forced with the target radial wind profiles ($U_T$) that have three shear strengths with the IPLs approximately located at 200 m (Fig. 1a). Similarly, group H simulations also have three shear strengths with the IPLs between 400 and 500 m. The target tangential wind profile ($V_T$) is specified in Fig. 1b. The $V_T$ profile with the shear occurring below 700 m (dash-dotted) is used for group L simulations, the other (solid) for group H. This paper is focused on the radial wind shear because of its direct link to the inflection instability (GG14). Therefore, only one target tangential wind is prescribed for each simulation group, which has three target radial wind profiles as discussed above. It is recognized that changes in the radial wind inevitably affect the tangential wind. The sensitivity of the LES results to the tangential winds is also explored. The simulations and relevant parameters are listed in Table 1.

While there is some quantitative difference between the target wind profiles defined above and the ones derived from the basic HBL balance equations such as those of Foster (2005), they carry some essential features that are similar to the model-derived or observed wind profiles such as an inflection point in the radial wind, the super-gradient wind in HBL, and the gradient wind balance above the HBL. Given our objective of investigating the impact of the wind shear (including both the shear strength and the inflection point level) on the roll structure, our choices of the target winds are justified in the sense that they retain the basic HBL mean wind features and provide a simple way to make a meaningful comparative study.

## 3 Overall turbulence structure

This section is centred on comparing instantaneous turbulence fields and statistics between groups L and H simulations (see Table 1). A special attention is given to the roll structure manifested by the coherent and organized turbulent flow. All the profiles presented here are obtained from ensemble averaging applied over the entire horizontal domain and between 8 and 10 h with a sample interval of 30 s. A time series of an average variable is constructed by taking the horizontal mean every minute.

**3.1 Time evolution and mean state**

To gain a general impression of the HBL development and differences among the simulations, the time series of the HBL heights ($z_i$) and mean profiles are examined. As shown in Fig. 2, $z_i$ increases rapidly with time for most simulations during the first 5 h, after which the growth slows down considerably, implying a quasi-equilibrium state being reached. H2 appears to be an exception; its HBL height becomes slowly growing only after 8 h. There is a clear tendency that stronger radial wind shear results in a higher $z_i$ for each group (L or H). The simulations L1 and H1 have the lowest $z_i$ in their group in accordance to the weakest turbulence likely due to the weak radial wind shear for both cases. It is worth noting that H3 predicts the highest $z_i$ among all the simulations, suggesting that it produces the strongest turbulence intensity even though it does not have the strongest radial wind shear (Table 1). It will be shown in Sect. 4 that H3 produces the most vigorous roll structure, which likely contributes to the highest $z_i$ through strong nonlocal mixing as discussed by GG14. In addition, $z_i$ has critical impact on the roll characteristics and their coupling with internal waves (GG14), which will be discussed in later sections. It also should be noted that the high $z_i$ may reflect the fact that neither radial advection nor dabatic heating is included in the heat balance. These processes may affect the growth of the mixed layer (Kepert et al., 2016).

For all the simulations, the parameter, $-z_i/L_{mo}$, where $L_{mo}$ is the Monin-Obukhov length, is between 0.075 and 0.12 (Table 1). These values are considerably smaller than values of 0.5–0.65 that represent the shear-buoyancy regime transition found by Moeng and Sullivan (1994) in their study of the shear and buoyancy driven boundary layers, implying that the shear production of turbulence is dominant in all the simulations. The maximum value of $-z_i/L_{mo}$ among all the simulations is 0.13, which is considerably less than the lower criterion $-z_i/L_{mo} = 1.5$ for the formation of buoyancy-shear driven roll structures (Glendening, 1996). Therefore, any roll structure resulting from these simulations should not be explained by the buoyancy-shear mechanism.

Because the mean wind profiles are nudged toward the target winds, the last hour average winds exhibit the characteristics that bear resemblance to the target wind profiles (Figs. 1 and 2). For instance, the radial shear increases with the radial wind speed within each group. Group L has stronger radial wind shears and lower IPLs than group H. The mean tangential winds are very similar within each group. The mean potential temperature ($\bar{\theta}$) profiles show considerable variations because of different entrainment rates primarily determined by the shear generated turbulence as well as the surface heat flux.

**3.2 Roll visualization**

Two major differences in the wind forcing among the simulations are associated with the radial wind shear strength and the IPLs. How do these differences affect the roll structure as well as turbulence in general? The link between the wind shear profiles and flow pattern is evident in the horizontal cross sections of $w'$ at three levels, $z/z_i \in$ (0.2, 0.4, 0.9), from the two groups of simulations shown in Fig. 3. These plan views demonstrate quasi-linear patterns defined by up and down motions for all the simulations except H1 for which the pattern is not clearly recognizable at $z/z_i = 0.4$ and 0.9, although a narrowly spaced and weak quasi-linear pattern is present at $z/z_i = 0.2$. The absence of the coherent structure from H1 is likely due to the weakest wind shear associated with the inflection point, which fails to generate strong turbulence to support the roll growth. The quasi-linear structures from the other 5 simulations have strong vertical coherence shown at three levels. Therefore, these flow patterns can be identified as "roll structure".

It is evident that the rolls appear stronger, in terms of the maximum $\|w'\|$, with the increases in the radial wind shear intensity within each group, i.e., from L1 to L3 or H1 to H3. For example, $\|w'\|_{max}$ at $z/z_i = 0.2$ from L1 is about 5 ms$^{-1}$ compared with 7 m s$^{-1}$ from L2 and 10 m s$^{-1}$ from L3. The increasing shear also leads to an increase in the roll horizontal scale within each group. The scale can be roughly estimated based on the number of the rolls. It is about 1 km, 2 km, and 2.5 km for L1, L2 and L3, respectively, and 3 km and 3. 6 km for H2 and H3, respectively. Different IPLs in the radial wind profiles have crucial impact on the roll structure. A comparison of $w'$ between the simulations of these two groups (i.e., L2 vs. H2 or L3 vs. H3) in Fig. 3 indicates that the horizontal scales of the rolls tend to be larger for group H (3 km for H2 and 3.6 km for H3) than group L (2 km for L2 and 2.5 km for L3) due to the higher IPLs in the former. It is noteworthy that H3, which has the highest IPL and moderately strong wind shear (Table 1), is characterized by the vigorous rolls that have the largest horizontal scale, implying the importance of the IPL in regulating the roll intensity as well as the scale. The roll expansion from these simulations is consistent with the general increase in the HBL heights, the enhanced wind shear, and the rising IPLs (Fig. 2).

These simulations also show strong signature of gravity waves. For example, the linear roll patterns are well defined near the inversion base (i.e., $z/z_i = 0.9$ in Fig. 2). These patterns even extend above the HBL (not shown here). The wave amplitude is particularly robust in H3. Strong evidence of gravity waves also comes from the turbulent statistics discussed in the next section. It is likely that this roll-like patterns within the inversion is connected to both the gravity waves and the roll structure in the HBL. The fact that H3 produces the strong rolls as well as the large gravity wave amplitude hints the possibility of an interaction between these two processes. This is consistent with previous studies such as GG14 and NN12, who found that internal gravity waves may be excited by the roll motion in the HBL and they interact with the rolls to enhance the associated turbulent transport.

Many of the above discussed aspects of the roll structure are also evident in the horizontal cross sections of other perturbation variables. Fig. 4 shows the wind component perturbations ($u'$ and $v'$) and their vertical fluxes ($w'u'$ and $w'v'$) at $z/z_i = 0.2$ from H3. The negative $v'$ tends to correlate with positive $w'$ along each narrow quasi-linear band. These negative $v'$ bands are caused by the upward motion transporting lower speed wind upward, directly resulting in a very similar roll pattern in the $w'v'$ field (Fig. 4b). These patterns suggest that the roll-scale tangential momentum flux is dominated by the downward transport (i.e., negative momentum flux) driven by the vigorous upward motion. The radial wind perturbations ($u'$) also show similar roll feature with the black line indicating a convergence line with $u' \sim 0$, corresponding to the positive $w'$ in H3 (Fig. 4a). It is interesting that the roll patterns are barely distinguishable in $w'u'$ (Fig. 4d), in contrast to $w'v'$ (Fig. 4b), although they are evident in both $w'$ and $u'$ fields (Fig. 4d). The poor correlation between $u'$ and $w'$ near the surface is likely due to the alignment of the roll axis, namely, nearly along the tangential direction. This can be seen by assuming the rolls are strictly two-dimensional and ignoring the small angle between the roll axis and the tangential direction. The continuity equation reduces to $\partial u'/\partial z + \partial w'/\partial z = 0$ and $w'$ can be written as $w' = -\int_0^z \partial u' / \partial x \cdot dz$, which implies that the vertical velocity near the surface is mainly driven by the low-level convergence of the radial flow. The above expression also implies that $u'$ and $w'$ are approximately $90^0$ out of phase: when $u'$ reaches a maximum or minimum, $\partial u' / \partial x \sim 0$ and, therefore, $w' \sim 0$. Similarly, when $w'$ reaches a maximum or minimum, $u' \sim 0$.

This argument is supported by further quantitative analysis. A coordinate transformation is performed on the instantaneous fields so that the resultant $u$ velocity is perpendicular to the longitudinal roll alignment while $v$ is along the longitudinal direction.

Then all the turbulent perturbations can be averaged over a distance (5 km in this case) along the roll direction to provide a snap shot of mean roll circulations on an $x$-$z$ cross section. Figure 5 shows the roll velocity perturbations and the layer averaged radial convergence ($\frac{1}{z}\int_0^{\bar{z}} \partial u' / \partial x \, dz$) at $z = 90$ m and $z = 500$ m. It is evident that $u' \cong 0$ at 90 m coincides with the strong convergence and positive $w'$ values near $x \sim 6.4$ km, 9.8 km, and 13.5 km indicated by the open circles, making positive $w'$ correlate with both positive and negative $u'$. At 500 m, however, the locations with $u' \cong 0$ move toward the rotation centre (i.e., toward the left), thus enabling a better correlation of positive $w'$ with negative $u'$. The cross section of the roll circulation from Fig. 6 shows that updrafts are originated along the convergence slope where $u' \cong 0$ and tend to coincide with negative $u'$ above the slope, leading to a downward (or negative) cross-roll momentum transport aloft. The negative momentum flux (i.e., $\overline{w'u'}$) in conjunction with the positive wind shear represents energy production for roll circulations. This result also agrees with those of Foster (2005) and GG14, which show that the roll streamlines tend to tilt vertically to efficiently extract the kinetic energy from the mean shear flow.

### 3.3 Turbulence statistics

Turbulence statistics respond strongly to the different wind profiles as demonstrated in Fig. 7. The negative radial momentum flux ($\overline{w'u'}$) is significantly enhanced with the increase in the radial shear intensity for group L or H, particularly near the levels of the inflection points, where the shear reaches its local maximum as pointed out by GG14. The higher IPLs from group H enhance $\overline{w'u'}$ in the upper portion of the HBL because of the increased shear layer depth (Fig. 2), which is also discussed by GG14. The stronger turbulence aloft in group H simulations further intensifies the entrainment across the inversion, leading to a deeper HBL. Both H3 and L3 have similar $\overline{w'u'}$ maxima in spite of the large difference in their shear shown in Table 1. The tangential momentum flux ($\overline{w'v'}$) also strengthens from group L to H responding, in part, due to the enhanced tangential wind shear above 300 m (Fig. 2). The $\overline{w'v'}$ of H3 increases the most since the turbulence (e.g. $\overline{w'^2}$) is considerably stronger in the upper part of the HBL than in other simulations. One consequence of the $\overline{w'v'}$ increase is to reduce the surface "friction" effect on the tangential wind speed because the overall HBL flux gradient is decreased as a result of the enhanced downward $\overline{w'v'}$ in the mid- and upper HBL. Recalling the robust roll structure from H3 (Fig. 3), we interpret the strengthening of turbulence as resulting from the highly organized and effective roll transport. This reasoning is supported by the spectral analysis presented in Sect. 4.

The buoyancy flux ($C_p \rho_0 \overline{w'\theta_v'}$) decreases from the fixed value 20 W m$^{-2}$ at the surface to the maximum negative entrainment flux at the inversion base (Fig. 7f). It is well documented that the ratio between the entrainment and surface heat flux is $-0.2$ for free convection generated by the surface heat flux (Stull, 1976; Conzemius and Fedorovich, 2006). Thus, the effect of wind shear on $C_p \rho_0 \overline{w'\theta_v'}$ is evident as the magnitude of this ratio can be as large as $\sim -1.5$ for H3. Variance of each wind component (i.e., $\overline{u'^2}$, $\overline{v'^2}$ or $\overline{w'^2}$) increases with the shear strength for both groups H and L (Figs. 7c–7e). The simulation H3, which has the highest inflection point and moderately intense shear, produces the strongest turbulence above 500 m. Above the HBL, neither $\overline{w'^2}$ nor $\overline{\theta'^2}$ is close to zero; they are in fact very large for L2, L3, H2 and H3. At the same levels, $\overline{w'\theta_v'}$ is very small or close to zero as shown in Figs. 7e–7g. This strongly suggests the presence of internal gravity waves above the HBL, which are

presumably generated by mesoscale perturbations associated with the HBL rolls. According to linear wave theory, there is a $90^0$ phase lag between the wave-induced vertical velocity and potential temperature perturbations, and therefore, the vertical heat flux associated with wave-induced perturbation is zero, although the vertical velocity and potential temperature variances can be large. The presence of gravity waves above boundary layer rolls is consistent with results from many studies including both LES (e.g. NN12) and 2-D model studies (e.g. GG14). The skewness of the HBL flow, defined by $S_w = \overline{w'^3} / \overline{w'^2}^{3/2}$ (Figs. 7h and 7i), represents the symmetry, or lack thereof, in the turbulence structure. That all the $S_w$ values above 150m are positive points to a positively skewed structure, that is, the flow is characterized with narrower/stronger updrafts and broader/weaker downdrafts (Zhu, 2008; Foster, 2005). In general, a high degree of the flow asymmetry is reached in the upper portion of the HBL.

Some important features emerging from the above diagnosis are worthy of emphasis: 1) all simulations except H1 produce well-defined roll structure manifested by a quasi-linear pattern through the depth of the HBLs; 2) increasing the vertical shear of the radial wind results in enhanced turbulence, higher HBL height, and larger roll spatial scales; 3) rising IPL also leads to a larger roll spatial scale in spite of the weakened radial shear; 4) the vertical tilting (in the radial direction) of the low-level convergence zone enhances the radial momentum flux associated with HBL roll circulations, which is consistent with other studies (e.g. GG14); and 5) the presence of internal gravity waves is strongly suggested by the "roll-like" pattern above the HBL and the 90 degree lag between $w'$ and $\theta'$ implied by the turbulence statistics. Some of these features are further confirmed by the spectrum analysis described in the next section.

## 4 Spectrum analysis

To understand how the turbulent flow at various scales respond to the changes in the wind forcing and how effective rolls are in vertical momentum transfer, we examine the 2D power density spectra of the simulated $w'$ and its co-spectra with $u'$, $v'$, and $w'^2$ at $z/z_i = 0.4$ where the rolls are most robust. The focus on 2D spectra instead of 1D is due to the fact that the former represents spectral peaks and associated spatial information more reliably than the latter as discussed by Kelly and Wyngaard (2006).

### 4.1 Turbulence spectra

All the spectra are calculated using the data collected between 8 and 10 h with a sampling interval of 5 min. They are functions of the magnitude of the horizontal wavenumber vector $k_h = \sqrt{k_x^2 + k_y^2}$, where $k_x$ and $k_y$ are the wavenumber in the radial and tangential direction, respectively. Note that the subscript "$x$" and "$y$" represents the radial and tangential direction, respectively, as defined in Sect. 2.1. Figures 8 and 9 compare various turbulence spectra at $z/z_i = 0.4$ among simulations within each group as well as between the two groups. For each group, the power of $w'$ increases with the enhancing wind shear at all wave numbers (Fig. 8a). This increase, however, is more significant for the wave numbers less than 0.01 m$^{-1}$, i.e., the spatial scales larger than 600 m, which is particularly true for group H. The changes in the spectral distribution from group L to H are more complicated because the higher IPLs are associated with weaker wind shear (Fig. 2). The major difference between the two groups occurs at the wavenumbers between $10^{-3}$ and $5 \times 10^{-3}$ m$^{-1}$. The H2 spectrum remains essentially the same as the L2 for the wavenumber $k_h \geq 0.008$ m$^{-1}$, below which the H2 power becomes lower than the L2 before it reaches the narrow peak at $k_h = 0.002$ m$^{-1}$. For L3 and H3, their spectra are very close to each other except that the latter (H3) exhibits a peak at a smaller wavenumber (i.e., $k_h = 0.0017$ m$^{-1}$, or wavelength ~ 3.6 km) than the former (L3) (i.e., $k_h = 0.0027$ m$^{-1}$, or the wavelength ~ 2.3 km). The spectral

peak from H3 is the strongest and its wavenumber is the smallest among all the simulations (Table 1). In contrast to the relatively smooth shape of the group L spectra, the spectra of both H2 and H3, which have higher IPLs than group L, exhibit a narrow peak (Fig. 8a), indicating the presence of a highly energetic and single-mode structure. This qualitative difference suggests that IPL plays a critical role in determining the roll strength and the effectiveness of the turbulent transport.

Many of the essential features discussed for the $w'$ power spectrum are also evident in the cospectrum of $w' - w'^2$, $w' - v'$, and $w' - u'$ in Fig. 8b and Fig. 9. Note that the covariance of $w'$ and $w'^2$ gives $w'^3$, which is related to the skewness. The cospectrum of $w' - w'^2$ from each of the simulations L2, L3, H2 and H3 is consistent with the corresponding $w'$ power spectrum in that both have the same peak wavelength. The cospectrum peak from H3 is the most prominent in that it is both large and narrow, implying that the roll vertical motion is strongly and positively skewed.

A major feature of the cospectrum of $w'$ and $v'$ from both H2 and H3 of Fig. 9 is its sharp negative maximum at the same peak wavenumber as that from the $w'$ spectrum, suggesting significant roll contributions to the longitudinal momentum flux. Compared with H2 and H3, the group L cospectra show a much smaller maximum even though their peak wavenumbers are the same as those of the rolls derived from the $w'$ spectrum. For $w' - u'$ cospectra, only H3 results in the same peak wavenumber as the rolls defined by the $w'$ spectrum, while other simulations produce the peak wavenumbers that are larger than the corresponding rolls. Therefore, the roll structure of H3 has the strongest spectral peaks, among all the simulations, at the same roll wavelength in the $w'$ power spectrum and the cospectra of $w' - w'^2$, $w' - v'$, and $w' - u'$.

The following features associated with H3 are worth noting: 1) the highest $z_i$ (Fig. 2); 2) the strongest turbulence intensity and momentum fluxes above 500 m (Fig. 7); 3) the largest roll wavelength (Fig. 8 and Table 1); and 4) the strongest peak at the roll wavelength of the turbulence power spectra and co-spectra among all the simulations (Fig. 8 and 9). These features suggest that H3 has produced the most robust roll structure because of the highest IPL in the radial wind and associated relatively strong shear (Fig. 2b).

It is also noteworthy that the presence of a significant narrow peak in the momentum flux spectra is consistent with the observational analysis by Zhang et al (2008), which shows sharp peaks in all the cospectra of $w'$ and the horizontal wind and temperature perturbations (their Fig. 9). A main difference is that their observed peak occurs at 900 m with an aspect ratio ~ 2 and our LES modelled is at 3.5 km at the ratio of 2.7.

**4.2 Spectral decomposition of turbulent fluxes**

How significant are the HBL roll contributions to turbulent fluxes compared to other turbulent eddies in the LES simulations? This issue has been addressed previously with a decomposition method based on the roll coherence feature. For example, the updraft-downdraft roll circulation can be defined based on the quasi-linear longitudinal coherence of the roll structure (Glendening, 1996); the roll-scale characteristics may also be represented as conditional means of the turbulent flow based on the convection model (Zhu, 2008). Because a key feature of the rolls is that turbulence is organized in such a way that various flux spectral distributions reach their maxima at the roll wavelength, a decomposition method based on spectral analysis provides a more fundamental representation of roll characteristics. This approach is also consistent with the observational analyses of HBL rolls by Zhang et al (2008).

To compute the contributions from different wavenumbers, we integrate each flux over three spectral bands to yield the subtotals at each model level. The spectral bands are chosen, in principle, to represent turbulent fluxes from the small scale, the large-eddy scale, and the roll scale based on the H3 spectra (Figs. 8 and 9). The small scale ranges from 0.1 km to 1 km; the large-eddy 1 km to 2.5 km; the roll 2.5 km to 12 km. The calculation is carried out from the surface to 2 km.

To emphasize the relative importance of the fluxes from the different spectral groups, we calculate both the fluxes and the flux fractions defined by the ratio of the specific group flux to the total, as shown in Fig. 10. The small-scale contribution to $\overline{w'^2}$ dominates in most of the HBL; the large roll variance increases significantly with height to the top of the HBL above which it carries more than 70 % of the total variance (Fig. 10a). This is consistent with the characteristics of the flux profiles implying the presence of gravity waves above the HBL (Fig. 7e–7g).

The longitudinal momentum fluxes ($\overline{w'v'}$) from different spectral groups exhibit different vertical distribution with the small-scale reaching the maximum near the surface and the larger scale near the mid-HBL (Fig. 10b). This difference reflects different nature of the turbulence at different scales. The small-scale turbulence is largely produced by the wind shear near surface, thus the flux maximum is naturally close to the surface. The roll circulation, caused by the inflection point instability, generates the momentum flux that depends on the wind shear in both the tangential and radial direction in the mid-HBL. The momentum flux $\overline{w'v'}$ obtains the largest roll fractional contribution 43 % at the mid-HBL among all fluxes (Fig. 10f). The combined roll and large-eddy fluxes account for 65 % of the total. The roll contribution to $\overline{w'u'}$ is only 25 %; it is considerably weaker than the contribution to $\overline{w'v'}$, a result in accordance with the previous discussion that the radial flux has less roll coherence than the longitudinal one. The roll contribution to $\overline{w'^3}$ reaches the maximum at $0.7z_i$, accounting for about 20 % of the total (Fig. 10d and h), while the combined roll and large-eddy contribution is about 45 %.

**4.3 Roll characteristics: correlation coefficients and skewness**

We have argued that the correlation between the roll-scale $w'$ and $u'$ is weaker than that between $w'$ and $v'$ because the low-level convergence is mainly driven by the radial wind component, thus leading to the diminished $u'$ in the area where the roll $w'$ reaches the maxima. This reasoning is based on both the instantaneous perturbation fields (Figs. 4-6) and the momentum related cospectra (Fig. 9). It is also supported by quantifying correlations of $w' - v'$ and $w' - u'$ from the roll contributions shown in Fig. 10. These correlation coefficients are shown in Fig. 11. The absolute value of the coefficient $C_{wv}^r$, defined by $C_{wv}^r = \overline{w'v'}^r / [\overline{w'^2}^r \cdot \overline{v'^2}^r]^{0.5}$ where superscript $r$ represents the roll contribution, is around 0.47 from 30 m to 500 m, then decrease to near zero at 1 km. In contrast, $\left| C_{wu}^r \right|$ increases from 0 near surface to 0.3 at 200 m, keeps nearly constant up to 900 m, and then gradually decreases to 0.2 at 1.5 km. The values of $\left| C_{wu}^r \right|$ are smaller than those of $\left| C_{wv}^r \right|$ below 600 m, indicating a weaker roll correlation of $w'$ with $u'$ than with $v'$. The increasing value of $\left| C_{wu}^r \right|$ with height in the lowest 300 m is also consistent with the understanding of the tilted convergence zone allowing for more efficient radial momentum transfer away from the surface (Fig. 6).

The results of the roll contribution to the third moment $\overline{w'^3}$ may be used to characterize the roll structure such as the roll skewness ($S_w^r$), which can be computed in the same fashion as the correlation coefficients from the roll contributions to $\overline{w'^2}$ and $\overline{w'^3}$. Because the simulated skewness is likely problematic near the surface due to the coarse resolution (Sullivan and Pattern 2011), only the profile above 200 m is plotted in Fig. 11. The skewness $S_w^r$ decreases from 1.6 at 200 m to 0.7 at 600 m and to 0 above the HBL, where gravity waves are likely present. This decreasing-with-height tendency agrees with the calculation of Zhu (2008). Therefore, the roll updraft fraction is generally less than 50 % and increases with height. In addition, the roll-scale skewness is close to zero above the HBL, indicating that the flow at these scales is symmetric. This characteristic is consistent with the linear theory of internal gravity waves. For a linear wave at a given level, the updrafts and downdrafts in a horizontal domain with period boundary conditions applied along the side walls should occupy approximately the same amount of the fractional coverage.

The spectral analysis in this section confirms that both the roll's horizontal scale and intensity are highly dependent on the shear and IPL in the radial wind profile. The stronger the radial wind shear is and the higher the IPL is, the stronger and larger the rolls are. More importantly, increasing IPL tends to produce a robust roll structure in the sense that a narrow and sharp peak is present in the $w'$ power spectrum and its wavelength is the same as the peak wavelengths from the co-spectra of $w' - w'^2$, $w' - v'$, and $w' - u'$. This is in contrast to the weaker rolls (e.g. H2) for which the peak wavelength from the $w' - u'$ is shorter than the others because of the weaker coherency between the roll-scale $w'$ and $u'$.

**4.4 Momentum transfer coefficients**

The momentum transfer coefficient, defined by the negative ratio of the momentum flux to the mean wind shear according to the $K$ theory (Stull, 1988, p. 204), plays a central role in the representation of HBL. It has been shown and argued that the roll generated momentum flux cannot be represented by the local transfer theory because of the "large-scale" nature of the roll circulations in terms of its horizontal and vertical scales as compared to $z_i$ (e.g. Foster, 2005; Zhu, 2008; and Gao and Ginis 2016). In this subsection, the issue of the transfer coefficient is briefly discussed using the results from the spectral analysis. Because the momentum fluxes have been decomposed into three spectral groups, it is convenient to compute the transfer coefficient for each group. By definition, the transfer coefficient for the radial momentum flux from each spectral group $K_u^i$ can be calculated by

$$K_u^i = -\frac{\overline{w'u'}^i}{\partial \overline{u}/\partial z},$$ (3)

where superscript $i \in (s,l,r)$ represents small-scale (<1 km), large-eddy-scale (1–2.5 km), and roll-scale (>2.5 km), respectively. The transfer coefficient for the tangential momentum flux $K_v^i$ is computed similarly. Because both the momentum fluxes (except for $\overline{w'u'}^r$) and the vertical gradient of the wind speed are very close to zero above 1400 m, all the values of the computed transfer coefficients are removed for $z \geq 1400$ m.

These transfer coefficients of both wind components are shown in Fig. 12. The values of $K_u^i$ change little with height from 200 m to 1.1 km, above which $K_u^r$ increases significantly because of both the finite values of $\overline{w'u'}^r$ and near-zero gradient of $\overline{u}$.

The non-zero $\overline{w'u'}^{i}$ above HBL is likely caused by the internal gravity waves which are connected to the roll structure and have the same wavelength as the rolls as discussed previously (also see GG14). The transfer coefficients $K_u^i$ are ill-defined around $z$ = 200 m because $\partial \overline{u}/\partial z \approx 0$. Unlike the nearly constant $K_u^i$, the tangential transfer coefficients $K_v^i$ increase with height from zero at surface to $\sim$ 150 m$^2$ s$^{-1}$ at 850 m. They then sharply increase near the HBL top where $\partial \overline{v}/\partial z \approx 0$, which results in both very large positive and negative values of $K_v^i$ because $\overline{w'v'}^{i}$ is always negative while $\partial \overline{v}/\partial z$ changes sign. This behaviour is contradictory to the downgradient transfer theory which assumes none negative $K_v^i$ (e.g. Stull, 1988, p. 108). This is similar to the result of the counter-gradient $\overline{w'v'}$ for the same reason from the two-dimensional roll model of Gao and Ginis (2016). The main difference is that their counter-gradient feature occurs in the mid-HBL where the momentum flux is significantly larger than that near the HBL top in our simulation H3 (Fig. 10b). This difference is mainly caused by the different mean tangential wind profiles obtained with different methods: the dynamic model approach of Gao and Ginis (2016) and the mean nudging in this work. Therefore, there is a need to apply the same mean wind profiles in both the 2-D roll and LES models for a more effective comparison. The sub-grid scale parameterized flux is not included in either $\overline{w'v'}^{i}$ or $\overline{w'u'}^{i}$. The inclusion of the SGS flux would slightly change the small-scale transfer coefficient profiles $K_v^s$ and $K_u^s$.

Overall, there are marked differences between $K_u^i$ and $K_v^i$ in the mid-HBL between 200 and 850 m. The values of either $K_u^i$ or $K_v^i$ do not vary greatly between the spectral groups even though the differences are obvious. The counter-gradient feature occurs at the HBL top where $\partial \overline{v}/\partial z$ changes sign and $\overline{w'v'}^{i}$ remains negative. Its effect on the momentum flux parameterization would be likely negligible in this case, because $\overline{w'v'}$ is very small near the HBL top.

**5 Impact of tangential wind shear**

We have so far emphasized the impact of the radial wind shear on both turbulence intensity and spectral distribution. However, both the radial and tangential winds may have significant shear above the surface layer (Fig. 2b and 2c). What roles does the tangential wind shear play in regulating the roll structure? This section attempts to address this issue by comparing the simulations H3, L3, L3H, and H3L, which are forced with different radial and tangential wind shear in the target profiles (Table 1). The simulation L3H uses the same target radial wind profile as the L3, but the same target tangential wind as the group H simulations (i.e. the profile H in Fig. 1). Correspondingly, the H3L adopts the same target radial wind profile as the H3, but the target tangential wind of the group L (i.e. the profile L in Fig. 1). This target wind specification is designed to examine how the roll structure responds to a change in one wind component while the other remains the same.

The comparison of the turbulence statistics profiles from H3 and H3L with those from L3 and L3H (Fig. 13) suggests that the radial wind plays a dominant role in determining the turbulence intensity. The target radial wind with a high IPL from H3 and H3L leads to both the stronger $\overline{w'u'}$ and higher HBL tops than the wind profile with a low IPL from L3 and L3H, regardless of different target tangential wind used. The tangential momentum flux $\overline{w'v'}$ is, however, predominately determined by the tangential wind shear (Fig. 13a). Both L3 and H3L result in similar weak momentum fluxes ($\overline{w'v'}$), which can be attributed to

the same target tangential wind profile L in Fig. 1b. The stronger momentum fluxes are obtained from H3 and L3H, which have the same extended higher-level tangential shear profile H in Fig. 1b.

The spectral response of the turbulence is displayed in Fig. 14. A dominant feature is that there is a peak in the power spectrum of $w'$ as well as the two co-spectra of $w'-v'$ and $w'-u'$ at the same wave number from H3 and H3L, which have the same radial wind with the higher IPLs (Fig. 1a). This is particularly true for the co-spectra of $w'-u'$. In contrast, the peak values of the spectra from L3 and L3H are more broadly distributed at higher wave numbers. It is worth noting that the peak in the $w'-v'$ co-spectrum of H3L is considerably weaker than that of H3 because of the tangential wind shear reduction at upper levels in the target wind profile L (Fig. 1b).

The above results suggest that the radial wind shear plays a more dominant role in determining the roll characteristics with regard to the scale selection, while the tangential wind shear strongly influences the tangential momentum flux $\overline{w'v'}$. Consequently, the tangential wind shear enhances the overall turbulence intensity, $e = \frac{1}{2}(\overline{u'^2} + \overline{v'^2} + \overline{w'^2})$, through the shear production. It can also affect the kinetic energy of roll circulations, $(\overline{u'^2} + \overline{w'^2})$, through the return-to-isotropy terms in the respective variance budget equations as shown in NN12. This result is largely consistent with the analysis of GG14 who found that the radial wind shear and IPL defines the roll characteristics regarding the mode selection and turbulence intensity. Their analysis, however, does not include contributions from the tangential wind shear to the roll energetics because of the two-dimensional nature of the dynamic model.

**6 Summary and conclusion**

A series of LES simulations have been conducted to examine the response of the roll structure to different mean wind shear conditions in terms of the radial wind shear strength and the IPL in an idealized HBL. A unique feature in our approach is that a mean wind nudging technique with specified target wind profiles is used to maintain the horizontal-domain average wind profiles without directly affecting turbulent perturbations. Two groups of simulations (L and H) are conducted. Each group uses the same target tangential wind profile, but the three radial wind profiles with different shear. Group H are designed to have higher IPLs ($\sim 430$m) in the radial wind than group L ($\sim 200$ m).

All simulations except H1, which has the weakest radial wind shear, produce the rolls manifested by a quasi-linear structure with the horizontal scale ranging from 1 km to 3.6 km. The roll structure extends from the near-surface level ($z/z_i \sim 0.1$) to the HBL top ($z/z_i \sim 0.9$). Within each group of simulations, increasing radial wind shear tends to enhance overall turbulence and increase the HBL height. Both the $w'$ power spectral peak and its wavelength increase with the enhanced radial wind shear, indicating that the shear regulates both the rolls' intensity and horizontal scale. Increasing IPLs, from group L to H, results in more vigorous rolls with distinctly narrow and sharp peaks in the power spectra. The most robust rolls are produced in H3, which is forced with the highest IPL and moderately strong shear in the radial wind. A unique and important feature of this roll structure is that the peak wavelength is the same among the power spectrum of $w'$ and the co-spectra of $w'-w'^2$, $w'-v'$, and $w'-u'$, implying that a consistent large roll contribution to all the relevant turbulent fluxes. This feature is in contrast to all other

simulations in which the peak wavelength from the $w' - u'$ is shorter than the others because of the weak coherency between the roll-scale $w'$ and $u'$ due to the quasi two-dimensionality of the roll structure.

One of the important features regarding the roll contribution to the vertical momentum flux is that the tangential wind is better correlated with the vertical motion than the radial wind in the lower half HBL. It is because the low-level convergence mainly comes from the radial wind, whose roll-scale perturbation is close to zero where the upward motion is maximized. The convergence zone is tilted with height toward the rotation centre to generate broader updrafts in the area of negative radial wind perturbations. Consequently, the negative correlation of upward motion and radial wind perturbation increases with height, which is supported by the roll momentum correlation coefficients calculated based on the spectral analysis.

Effects of tangential wind shear are also investigated. A sensitivity simulation, in which the upper level tangential wind shear is reduced, shows that the basic roll structure is not significantly impacted in the sense that both the power spectrum and the momentum flux co-spectra generally maintain their distributions. The tangential momentum flux, however, changes significantly with the tangential wind shear, which feedbacks to the turbulence generation and leads to some difference in the overall turbulence intensity. This effect is also reflected in the $w'$ power spectrum and tangential momentum flux cospectrum in which the peak values are reduced. Therefore, the radial wind profile critically determines the roll's presence, intensity, and scale, while the tangential wind shear has considerable impact on the tangential momentum transport.

The results of the spectral analysis are used to compute the roll contributions to various turbulent fluxes. The contribution from the roll-scale ($\geq 2.5$ km) circulation accounts for 15 % of $\overline{w'^2}$, 40 % of $\overline{w'v'}$, 20 % of $\overline{w'u'}$, and 20 % of $\overline{w'^3}$, respectively, at the mid-HBL. The corresponding large-eddy (1–25 km) contribution is 25 % ($\overline{w'^2}$), 30 % ($\overline{w'v'}$), 30 % ($\overline{w'u'}$), and 20 % ($\overline{w'^3}$), respectively. These values are, in general, consistent with previous studies (e.g. Zhu, 2008; Zhang et al., 2008). Because the magnitude of the negative roll tangential flux increases from almost zero to the maximum near the mid-HBL, the roll circulations tend to enhance the lower-level mean tangential wind by upward transport of the weaker wind. Finally, the momentum transfer coefficients derived from the three spectral groups show large differences between the radial and tangential components. While the counter-gradient behaviour occurs at the HBL top where the tangential wind maximum is reached, its effect is small as the momentum flux is almost negligible there in the case of H3. This evaluation based on Eq. (3) is meant to provide an example of the transfer coefficients. More in-depth analyses are clearly needed to understand the nature of the turbulent transfer organized by HBL rolls and develop new turbulence closure models for HBL.

This study highlights the critical roles of the radial wind shear in regulating the roll structure. As discussed in the introduction, the mean wind shear should be a strong function of both the local rotational forcing and the mesoscale tendencies. The mean nudging approach used in this work is intended to bridge the gap between the commonly used LES configuration and the need for including the mesoscale effects, and to facilitate sensitivity simulations. Because of the strong nudging it is difficult to isolate the impact of the rolls on the mean wind profile in this study. A more comprehensive study of the roll structure requires incorporating effects of the hurricane mesoscale environment such as radial wind advection. The LES approach recently proposed by Bryan et al. (2016) and the nested LES in a mesoscale model of Zhu (2008) provide attractive modelling frameworks that can be used to address issues related to the feedback of the rolls to the mean wind profiles in HBLs.

## Acknowledgement

We thank Dr James Doyle for discussions on the LES model setup and gravity waves. The careful reviews and valuable comments by Drs Kun Gao and Ralph Foster greatly improved the clarity of the manuscript. This research was funded by the Office of Naval Research (ONR) under program element (PE) 0602435N.

**Appendix:** Mean wind nudging

The mean wind nudging method introduced in Sect. 2 is used to maintain LES simulated mean wind profiles and to make systematic changes in the mean wind for sensitivity simulations; it has no direct influence on the resolved turbulence. Three LES simulations are presented here to evaluate these statements. The first simulation (RN1) uses the horizontal momentum equations with the rotation terms (i.e., the square bracket terms with $R = 44$ km) and without the nudging terms in Eq. (1) and Eq. (2). The second (RN2) keeps both the rotation and the nudging terms for which the target wind profiles are the same as the 9-10 h averaged wind from RN1. The third (RN3) removes the rotation term and keeps the nudging, and the target profiles are the same as those from RN2 except the target radial wind is enhanced to $-16.5$ m s$^{-1}$ at 90 m as shown in Fig. A1a. The relaxation time scale is 10 minutes.

In general, all the variables are in excellent agreement among the three simulations as shown in Figs. A1−A2. The simulation RN1 and RN2 have very consistent $z_i$ after 4 simulation hours, while the RN3 predicts $z_i$ that is 50 m lower than the others. The radial wind velocity at 60 m from RN1 oscillates around the mean value $-9.5$ m s$^{-1}$ after 1 h, which is consistent with that from RN2 and only 0.6 m s$^{-1}$ stronger than RN3 that excludes the rotation term. The significantly reduced oscillation in RN2 is due to the strong nudging, and the absence of the oscillation in RN3 reflects the removal of the rotation term. Despite these differences, all the mean and turbulence profiles compare well among these simulations. RN1 and RN2 almost have identical results as seen from Fig. A2. RN3 predicts slightly weaker turbulence in the upper HBL, being consistent with the weaker shear in both $\bar{u}$ and $\bar{v}$ at these levels. These results confirm the previous two-dimensional model simulations and LES analyses that the rotation terms do not have major influence on the turbulence structure driven by the wind shear, although these terms may play the dominant role in the case of the parallel instability (Foster, 2005, NN12). They also demonstrate that the mean wind nudging method can be used to examine the response of turbulence to a specific mean wind profile that is strongly regulated by the nudging process. All the simulations presented in Sects. 1-6 of this paper exclude the rotation terms and keep the nudging terms in Eq. (1) and (2) with the relaxation time scale 10 min.

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

**Table 1**: Simulation conditions and results with the following parameters: Individual experiments (Exp), maximum radial wind shear (RSH-$_{max}$), inflection-point level (IPL), target radial wind ($U_T$), target tangential wind($V_T$), wavelength at the peak of $w'$ power spectrum ($L_p$), HBL height ($z_i$), aspect ratio ($L_p/z_i$), and the ratio $-z_i/L_{mo}$, where $L_{mo}$ is the Monin-Obukhov length.

| Exp | RSH$_{max}$ | IPL | $U_T$ | $V_T$ | $L_p$ | $z_i$ | $L_p/z_i$ | $-z_i/L_{mo}$ |
| --- | --- | --- | --- | --- | --- | --- | --- | --- |
| | s$^{-1}$ | m | Fig. 1a | Fig. 1b | km | km | | |
| L1 | 0.0139 | 210 | L1 | L | 1.20 | 0.88 | 1.4 | 0.07 |
| L2 | 0.0216 | 210 | L2 | L | 1.94 | 1.05 | 1.8 | 0.08 |
| L3 | 0.0273 | 180 | L3 | L | 2.29 | 1.14 | 2 | 0.08 |
| H1 | 0.0053 | 480 | H1 | H | 1.01 | 0.91 | 1.1 | 0.08 |
| H2 | 0.0110 | 360 | H2 | H | 3.15 | 1.12 | 2.8 | 0.1 |
| H3 | 0.0142 | 450 | H3 | H | 3.60 | 1.35 | 2.7 | 0.13 |
| L3H | 0.0276 | 210 | L3 | H | 2.3 | 1.22 | 1.9 | 0.08 |
| H3L | 0.0147 | 420 | H3 | L | 3.1 | 1.25 | 2.5 | 0.1 |

**Figure Captions**

**Figure 1**: Target wind profiles used in simulations of groups L and H. (a) Target radial wind profiles; (b) target tangential wind profiles. Only one target tangential profile $V_T$ is used for each group.

**Figure 2**: HBL height evolution and mean vertical profiles. (a) HBL height $z_i$; (b) mean radial wind $\bar{u}$ ; (c) mean tangential wind $\bar{v}$ ; and (d) mean potential temperature $\bar{\theta}$ . Black dots in Fig. 2b denote the inflection point locations.

**Figure 3**: Plan view of $w'$ at 9 h at three different levels (i.e., $z/zi$ = 0.2, 0.4, and 0.9, respectively) from groups L and H simulations.

**Figure 4**: Plan views of turbulent perturbations at 9 h from H3 at $z/z_i = 0.2$. The fields are (a) $v'$ , (b) $w'v'$, (c) $u'$ , and (d) $w'u'$ . An "eye-fit" black line is drawn in (c) to show an example of convergence zone induced by the radial wind.

**Figure 5**: Phase differences between the along-roll averaged perturbation $u'$ , $w'$ , and the vertically integrated divergence, $div = \dfrac{1}{z}\displaystyle\int_0^z \partial u' / \partial x \, dz$ at $z$ = 90 m (top) and $z$ = 500 m (bottom) for H3. Open circles denote the locations where $u' = 0$ and $\partial u' / \partial x < 0$. Note the different vertical scales between the top and bottom panels.

**Figure 6**: Vertical cross-section of along-roll averaged perturbations from H3. The cross-roll velocity $u'$ is shown by colour shading. The value of $\partial u'/\partial x = -5\times10^{-3}$ s$^{-1}$ is contoured by thick black lines. Flow vectors are also displayed.

**Figure 7**: Profiles of LES turbulence statistics. The variables are (a) $\overline{w'u'}$ , (b) $\overline{w'v'}$, (c) $\overline{u'^2}$ , (d) $\overline{v'^2}$ , (e) $\overline{w'^2}$ , (f) $C_p\rho_0\overline{w'\theta'}$ , (g) $\overline{\theta'^2}$ , (h) $\overline{w'^3}$ , and (i) $S_w = \overline{w'^3}/(\overline{w'^2})^{3/2}$ .

**Figure 8**: 2-D power spectra of $w'$ (a) and co-spectra of $w' - w'^2$ (b) at $z/z_i = 0.4$

**Figure 9**: 2-D co-spectra of $w' - v'$ (a) and $w' - u'$ (b) at $z/z_i = 0.4$ .

**Figure 10**: Decomposition of turbulent fluxes for H3. Various spectral components for turbulent flux profiles are presented in top panels; fractional contributions from the components in bottom panels. Three spectral groups are small scale (< 1 km), large-eddy (1–2.5 km), and roll (> 2.5 km), respectively. (a) $\overline{w'^2}$ , (b) $\overline{w'v'}$ , (c) $\overline{w'u'}$, (d) $\overline{w'^3}$ , (e) spectral fractional contribution to $\overline{w'^2}$ , (f) contribution to $\overline{w'v'}$ , (g) contribution to $\overline{w'u'}$ , and (h) contribution to $\overline{w'^3}$ .

**Figure 11**: Vertical profiles of roll characteristics derived from H3: $w$ skewness ($S_w^r$); $w' - v'$ correlation coefficient ($C_{wv}^r$); and $w' - u'$ correlation coefficient ($C_{wu}^r$).

**Figure 12**: Momentum transfer coefficients for three spectral groups of H3 for $K_u$ (a) and $K_v$ (b). The three spectral groups are small scale (< 1 km), large-eddy (1–2.5 km), and roll (> 2.5 km), respectively.

**Figure 13**: Comparison among simulations H3, L3, H3L and L3H with different wind shear. (a) $\overline{w'v'}$; (b) $\overline{w'u'}$; (c) $\overline{w'^2}$ ; and (d) $\bar{\theta}$ .

**Figure 14**: Comparison of the power spectra of $w'$ (left), co-spectra of $w' - v'$ (centre) and $w' - u'$ (right) at $z/z_i = 0.4$ among L3, L3H, H3, and H3L.

**Figure A1**: Comparison of the evolution of the boundary layer height $z_i$ (a) and the radial wind component $\bar{u}$ at the 60 m level (b) from three tests RN1 (only rotation included), RN2 (both rotation and nudging included), and RN3 (only nudging included).

**Figure A2**: Comparison of test simulations for the mean nudging approach. All the profiles are averages between 9 and 10 h at a sampling interval 30 s. (a) Mean radial wind, (b) mean tangential wind, (c) $\bar{\theta}$ , (d) $\overline{w'^2}$ , (e) $\overline{u'^2}$ , (f) $\overline{v'^2}$ , (g) $\overline{w'u'}$ , (h) $\overline{w'v'}$ .

**Table Captions**

**Table 1**: Simulation conditions and results with the following parameters: Individual experiments (Exp), maximum radial wind shear (RSH$_{max}$), inflection-point level (IPL), target radial wind ($U_T$), target tangential wind($V_T$), wavelength at the peak of $w'$ power spectrum ($L_p$), HBL height ($z_i$), aspect ratio ($L_p/z_i$), and the ratio $-z_i/L_{mo}$, where $L_{mo}$ is the Monin-Obukhov length.

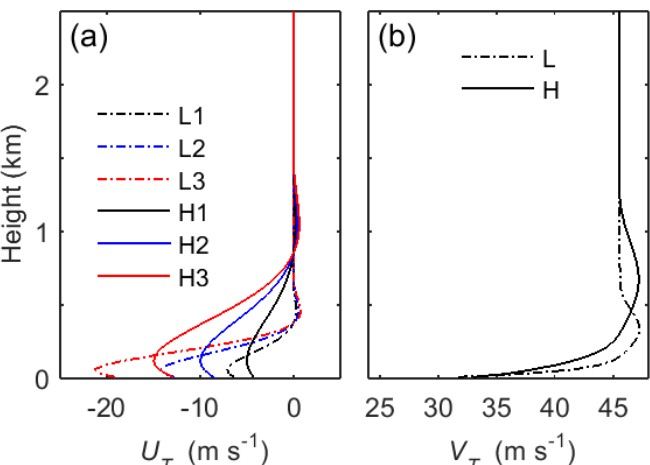

**Figure 1**: Target wind profiles used in simulations of groups L and H. (a) Target radial wind profiles; (b) target tangential wind profiles. Only one target tangential profile $V_T$ is used for each group.

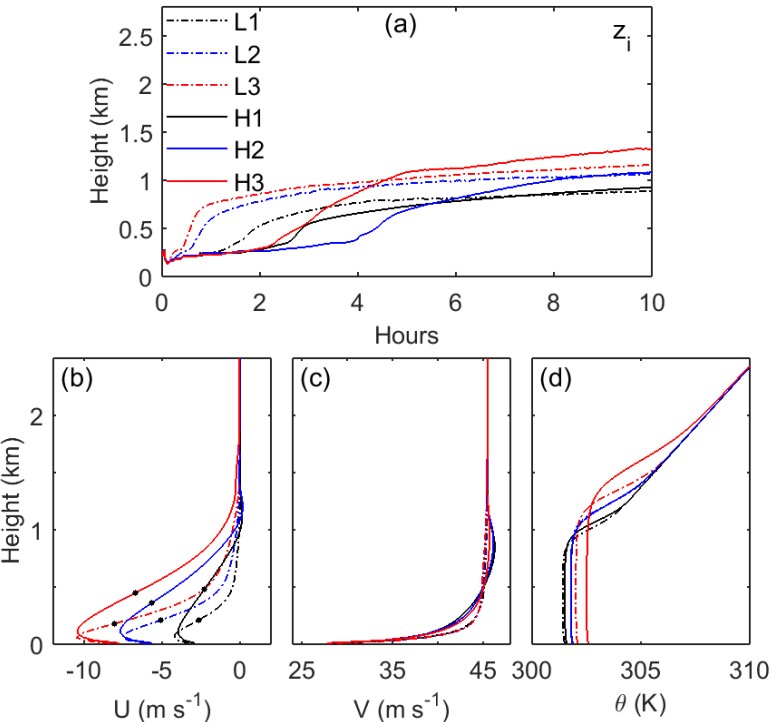

**Figure 2**: HBL height evolution and mean vertical profiles. (a) HBL height $z_i$; (b) mean radial wind $\overline{u}$ ; (c) mean tangential wind $\overline{v}$ ; and (d) mean potential temperature $\overline{\theta}$ . Black dots in Fig. 2b denote the inflection point locations.

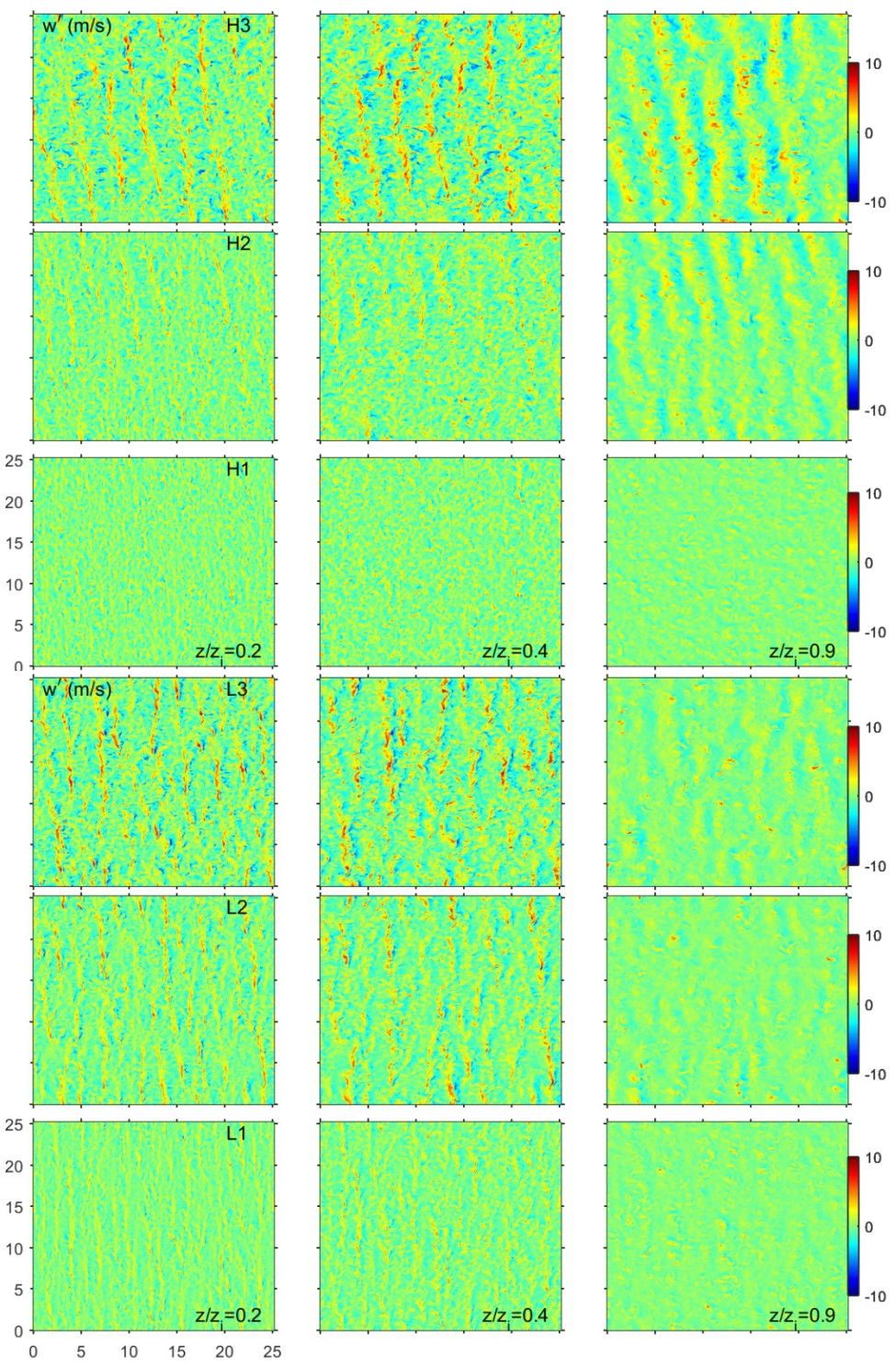

**Figure 3**: Plan view of $w'$ at 9 h at three different levels (i.e., $z/zi$ = 0.2, 0.4, and 0.9, respectively) from groups L and H simulations.

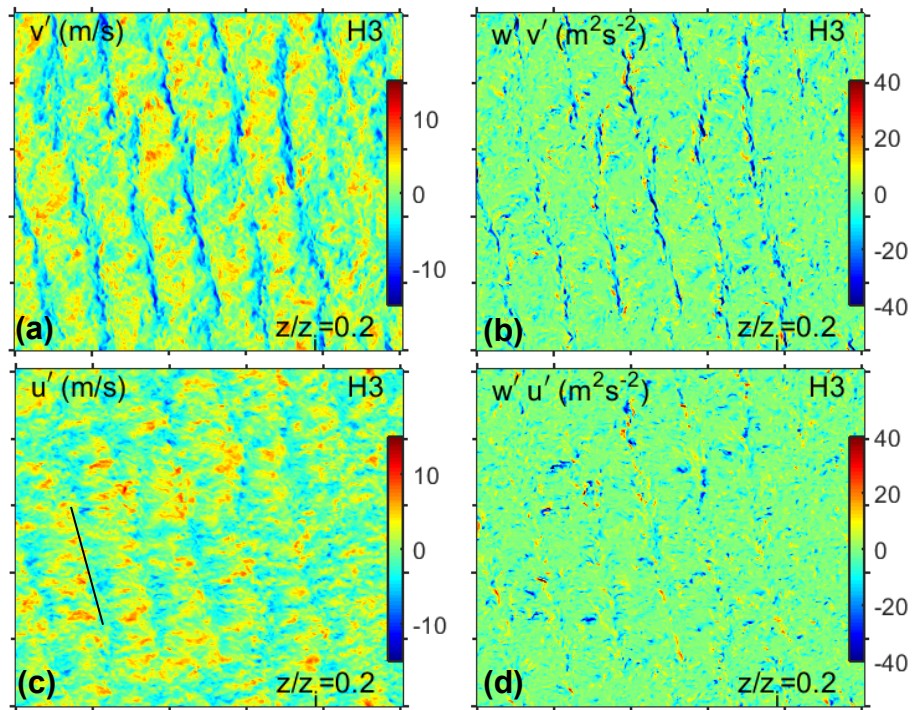

**Figure 4**: Plan views of turbulent perturbations at 9 h from H3 at $z/z_i$ = 0.2. The fields are (a) $v'$, (b) $w'v'$, (c) $u'$, and (d) $w'u'$. An "eye-fit" black line is drawn in (c) to show an example of convergence zone induced by the radial wind.

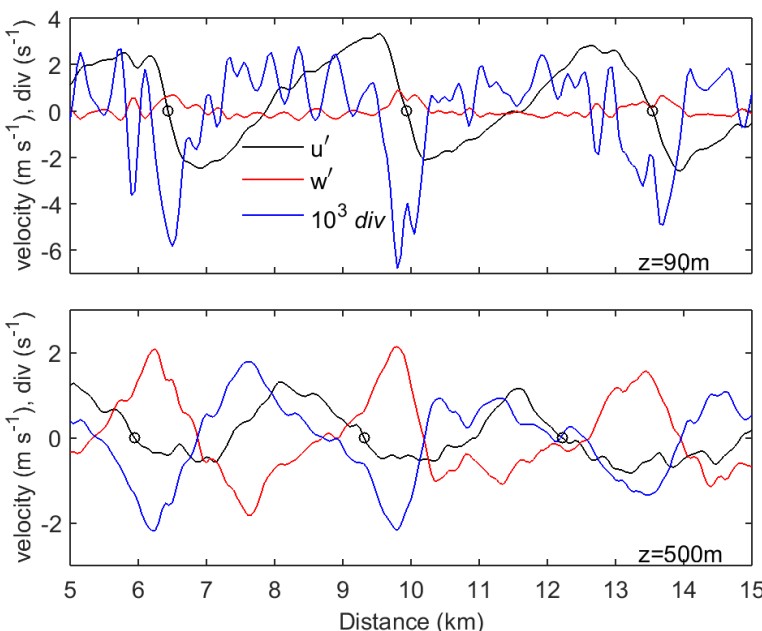

**Figure 5**: Phase differences between along-roll averaged perturbation $u'$, $w'$, and the vertically integrated divergence, $div = \dfrac{1}{z}\displaystyle\int_0^z \partial u' / \partial x \, dz$ at $z$ = 90 m (top panel) and $z$ = 500 m (bottom) for H3. Open circles denote the locations where $u' = 0$ and $\partial u' / \partial x < 0$. Note the different vertical scales between the top and

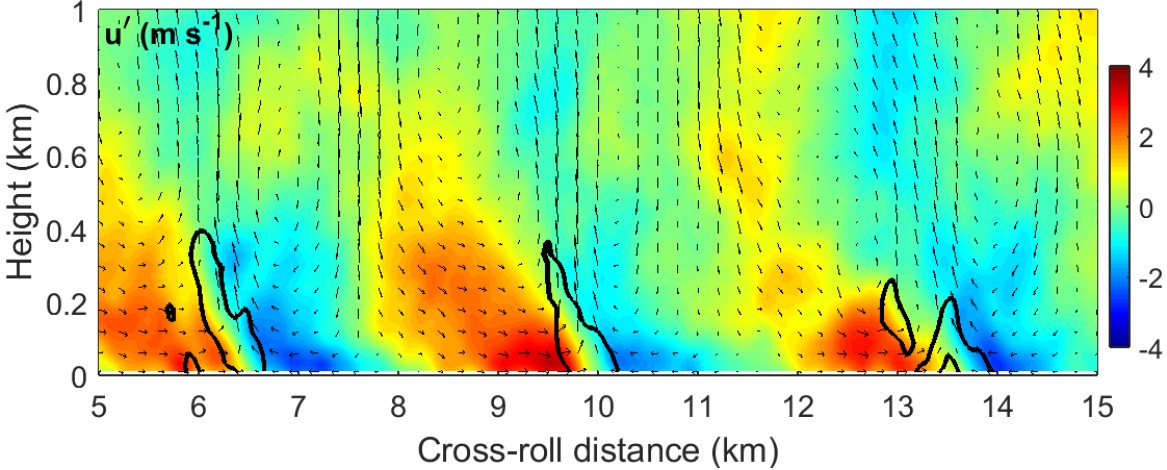

**Figure 6**: Vertical cross-section of along-roll averaged perturbations from H3. The cross-roll velocity $u'$ is shown by colour shading. The value of $\partial u'/\partial x = -5\times10^{-3}\,\text{s}^{-1}$ is contoured by thick black lines. The flow vectors are also displayed.

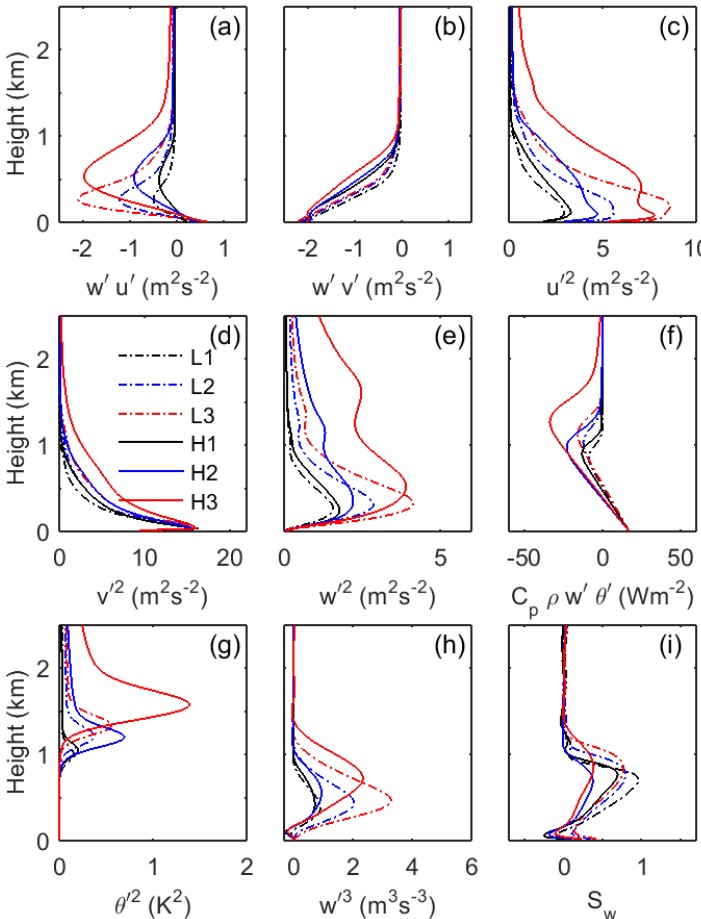

Figure 7: Profiles of LES turbulence statistics. The variables are (a) $\overline{w'u'}$, (b) $\overline{w'v'}$, (c) $\overline{u'^2}$, (d) $\overline{v'^2}$, (e) $\overline{w'^2}$, (f) $C_p \rho \overline{w'\theta'}$, (g) $\overline{\theta'^2}$, (h) $\overline{w'^3}$, and (i) $S_w = \overline{w'^3} / (\overline{w'^2})^{3/2}$.

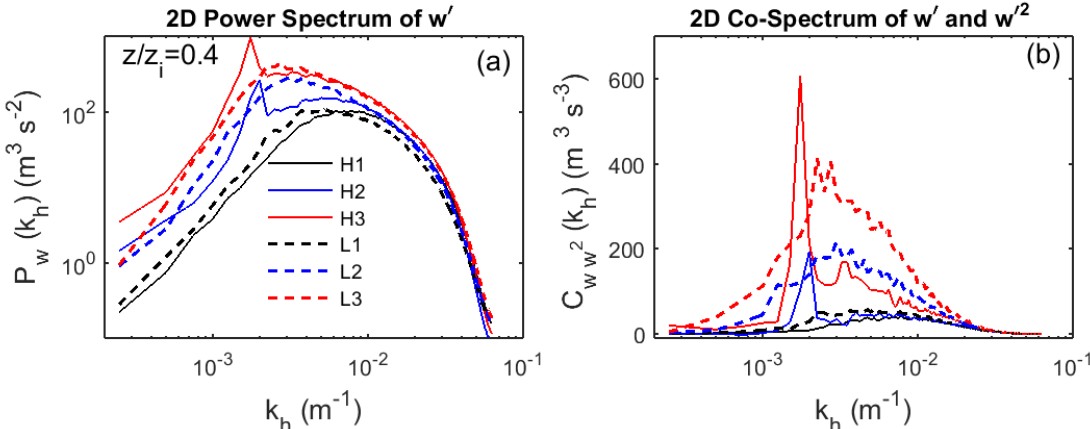

**Figure 8**: 2-D power spectra of $w'$ (a) and co-spectra of $w' - w'^2$ (b) at $z / z_i = 0.4$

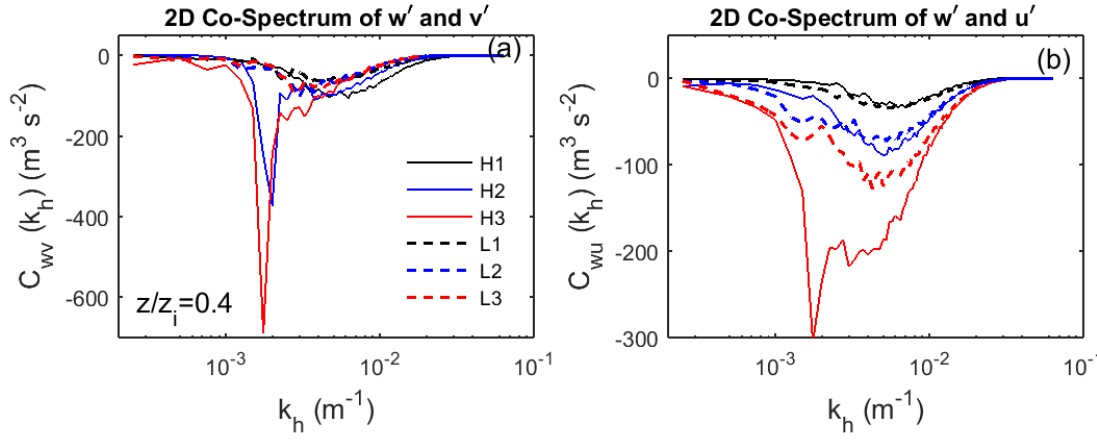

**Figure 9**: 2-D co-spectra of $w' - v'$ (a) and $w' - u'$ (b) at $z / z_i = 0.4$

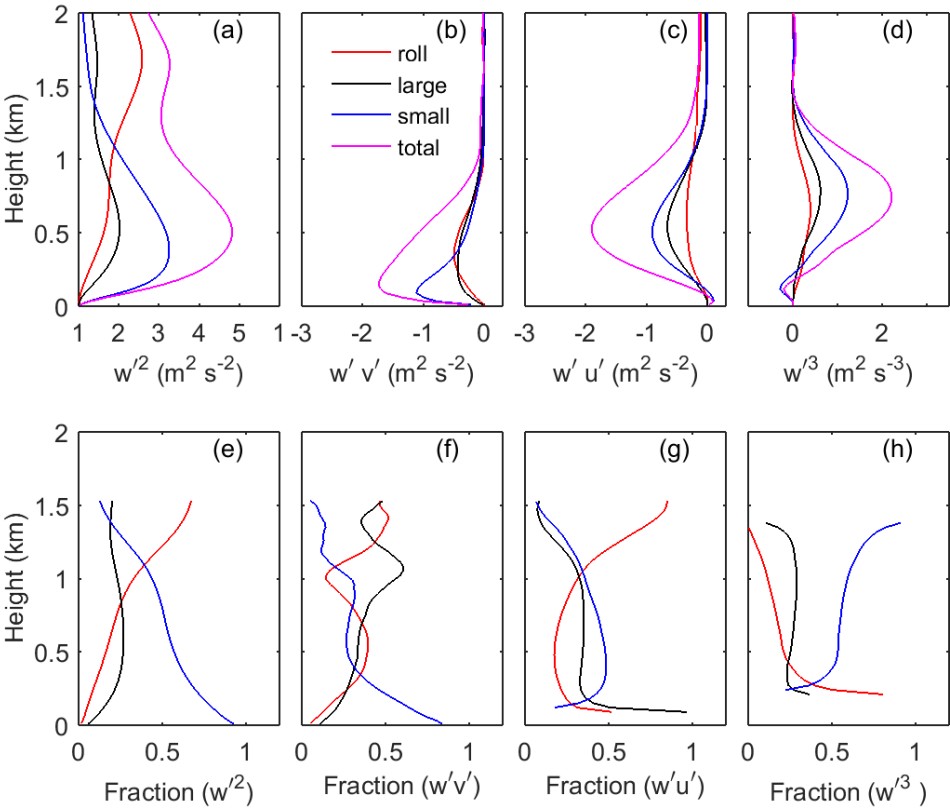

**Figure 10**: Decomposition of turbulent fluxes for H3. Various spectral components for turbulent flux profiles are presented in top panels; fractional contributions from the components in bottom panels. The three spectral groups are small scale ($< 1$ km), large-eddy (1–2.5 km), and roll ($> 2.5$ km), respectively. (a) $\overline{w'^2}$, (b) $\overline{w'v'}$, (c) $\overline{w'u'}$, (d) $\overline{w'^3}$, (e) spectral fractional contribution to $\overline{w'^2}$, (f) contribution to $\overline{w'v'}$, (g) contribution to $\overline{w'u'}$, and (h) contribution to $\overline{w'^3}$.

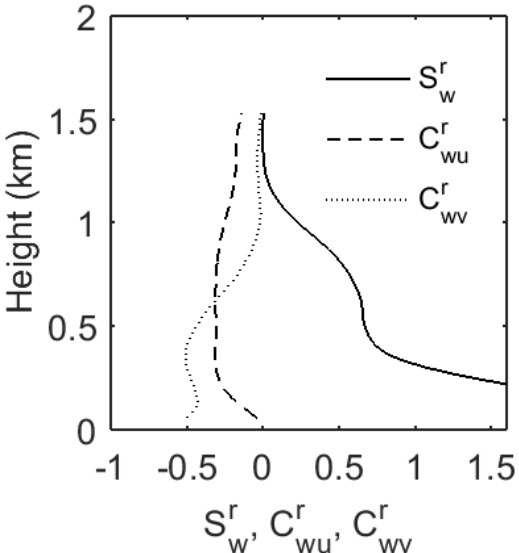

**Figure 11**: Vertical profiles of roll characteristics derived from H3: $w$ skewness ($S_w^r$); $w' - v'$ correlation coefficient ($C_{wv}^r$); and $w' - u'$ correlation coefficient ($C_{wu}^r$).

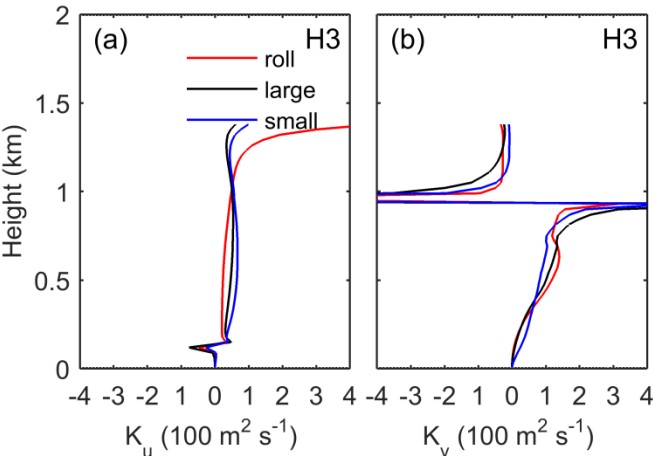

**Figure 12**: Momentum transfer coefficients for three spectral groups of H3 for $K_u$ (a) and $K_v$ (b). The three spectral groups are small scale (< 1 km), large-eddy (1–2.5 km), and roll (> 2.5 km), respectively.

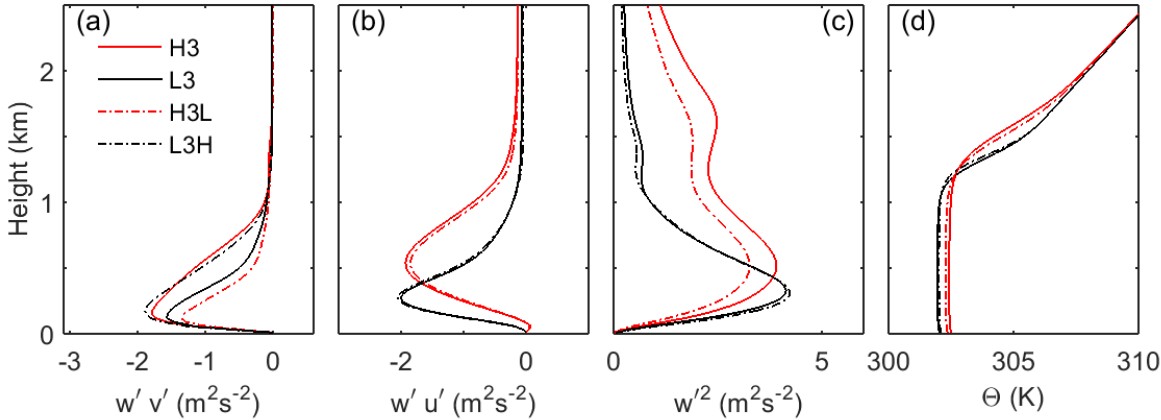

**Figure 13**: Comparison among simulations H3, L3, H3L and L3H with different wind shear. (a) $\overline{w'v'}$, (b) $\overline{w'u'}$, (c) $\overline{w'^2}$, and (d) $\overline{\theta}$ .

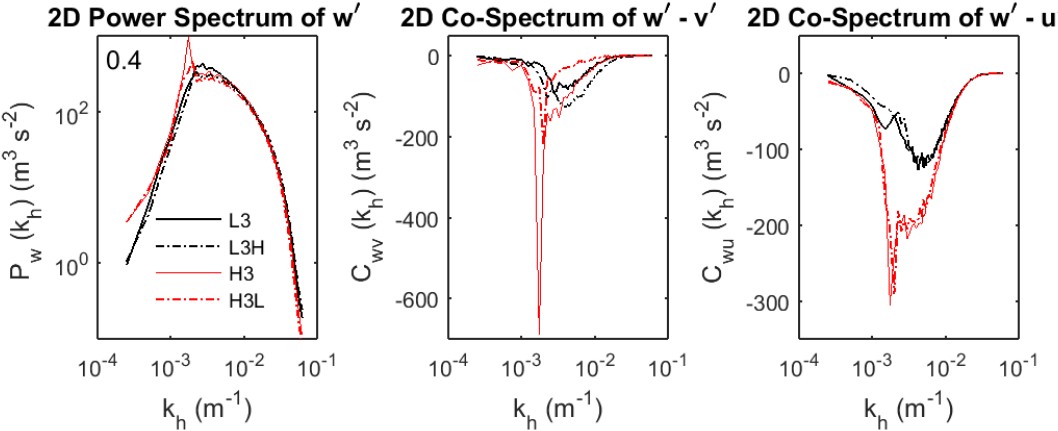

**Figure 14**:  Comparison of the power spectra of $w'$ (left), cospectra of $w' - v'$ (centre) and $w' - u'$ (right) at $z / z_i = 0.4$ among L3, L3H, H3, and H3L.

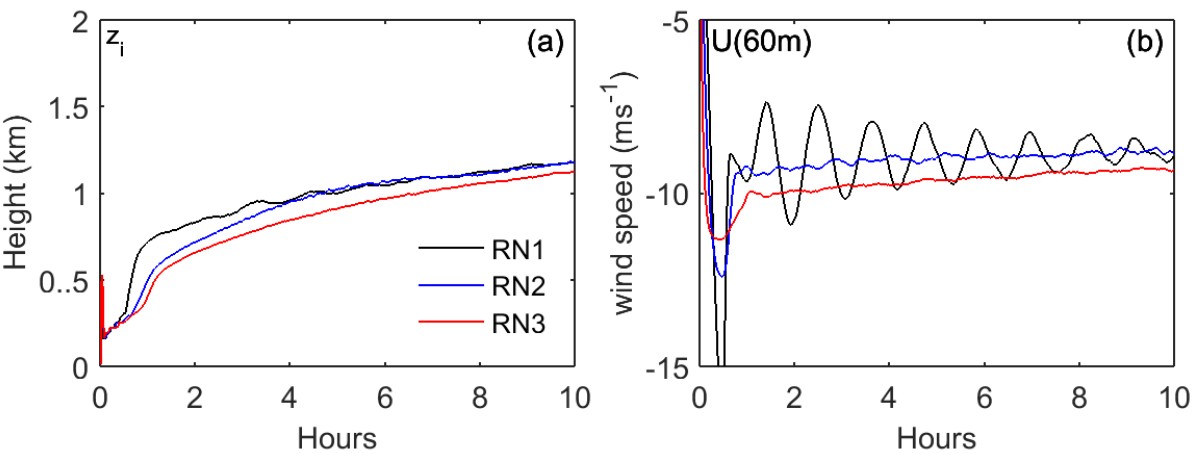

**Figure A1**: Comparison of the evolution of boundary layer height $z_i$ (a) and the radial wind component $\overline{u}$ at the 60 m height (b) from three tests RN1 (only rotation included), RN2 (both rotation and nudging included), and RN3 (only nudging included).

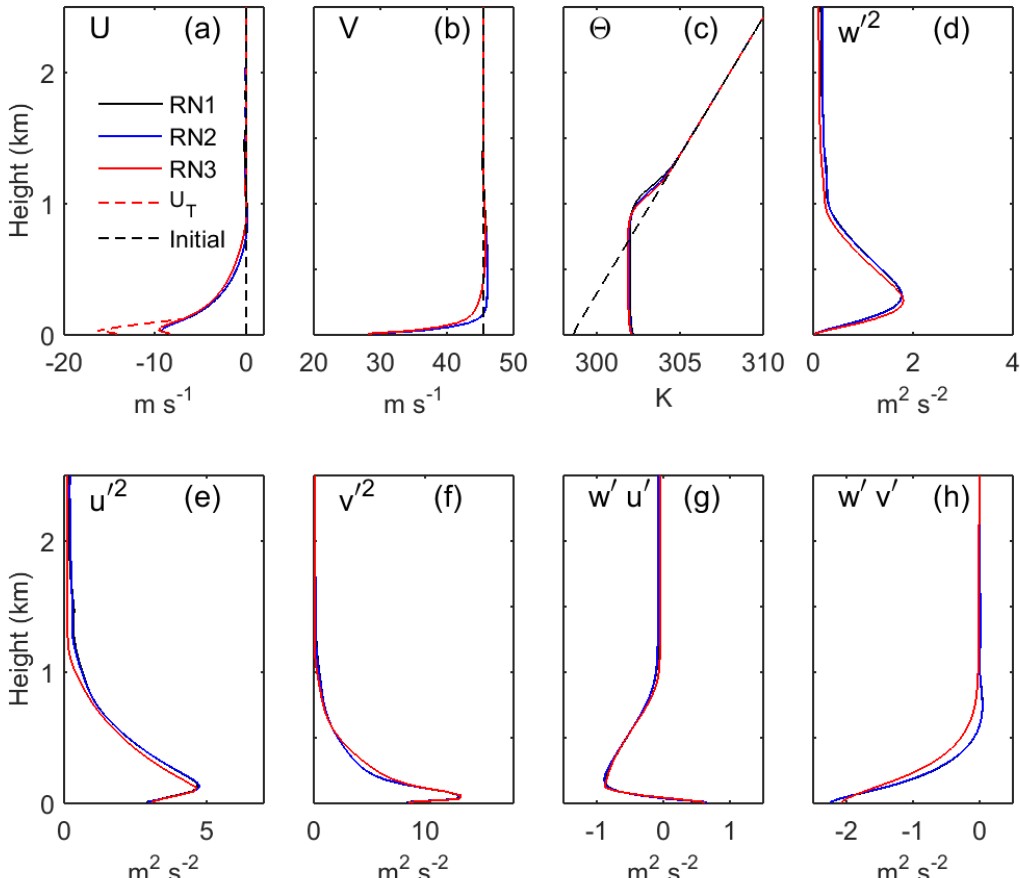

**Figure A2**: Comparison of test simulations for the mean nudging approach. All the profiles are averages between 9 h and 10 h at a sampling interval 30 s. (a) Mean radial wind; (b) mean tangential wind; (c) $\overline{\theta}$; (d) $\overline{w'^2}$; (e) $\overline{u'^2}$; (f) $\overline{v'^2}$; (g) $\overline{w'u'}$; (h) $\overline{w'v'}$.