# Peer review of "Impact of vertical wind shear on roll structure in idealized hurricane boundary layers"

_Atmospheric Chemistry and Physics, 2016_

## Referee Comment (RC1) · K. Gao (Referee) · 21 Nov 2016

This manuscript presents a well-designed and well-conducted study, which investigated the characteristics of rolls (large eddies) in the hurricane boundary layer under various sheared flow scenarios based on Large Eddy Simulations. The authors constrained the simulated mean (horizontal-averaged) flow by nudging it toward the prescribed wind profiles, making sure the simulated mean flow has characteristics consistent with observations. This study highlighted the importance of the radial wind shear in affecting the turbulence characteristics. Besides supporting some previous findings, the authors also found a few aspects of rolls that were not reported previously, such as the impacts of radial shear strength on the roll horizontal scale and the inflection

point height on the turbulent spectra. This manuscript is overall well written and the key points are clear. I would therefore recommend the manuscript for publication after the authors address the following comments.

Main comments

1. A few comments on the target wind profiles

i) The authors fixed the target tangential wind and only varied the radial wind in the two groups of experiments to explore the impact of the radial wind shear. However, in physics-based models one cannot vary the radial wind without changing the tangential wind as the two wind components are intrinsically coupled. Since the authors aimed to explore the turbulence characteristics in an idealized setting, I think their purpose justified their choice. But the authors are recommended to explicitly state this in their experiment design.

ii) There is no detailed description for how the target wind profiles are obtained. Are these profiles derived from a dynamical model or observed profiles?

iii) The authors only showed the LES results under the gradient wind speed of 45.5 m/s. Are the results shown in this manuscript representative for a range of gradient wind speeds that the authors have investigated? Or is the choice of 45.5 m/s somehow arbitrary? Either way, the authors should make it clear.

2. The model produced much higher mixed layer depth (Fig. 2a) than typically observed (Zhang et al., 2011, MWR). This is very likely because some important processes that stabilize the hurricane boundary layer, such as radial advection and diabatic effect (Kepert et al., 2016, JAS), were not considered in this study. The unusual high mixed layer has important implication for the large eddy characteristics. As discussed by Gao and Ginis (2014 and 2016), the height of mixed layer has critical impact on the roll characteristics and their coupling with internal waves. The authors are recommended to add some discussion on this.
3. The authors did nice quantitative analysis showing that the simulated rolls have a quasi-two-dimensional structure with the two velocity components (u', w') of the overturning circulations 90 degree out of phase. While their analysis help understand why the correlation between u' and w' is poor and the cross-roll momentum flux w'u' is weak, there is still lack of a fundamental explanation for the vertical tilting of the convergence zone (Fig. 6). One possible explanation is that, as shown in Foster (2005) and Gao and Ginis (2014), the roll streamlines tend to tilt vertically to efficiently extract the kinetic energy from the mean shear flow. The tilted convergence zone and the negative cross-roll momentum flux result from the tilted roll streamlines. The authors are suggested adding some discussion on this and revising the 4th point in the summary of section 3 accordingly.

4. The LES results are of central importance for the development of turbulent flux parameterizations under conditions where in-situ observations are difficult to obtain. This work presents important LES results under high wind shear conditions. The authors are thus encouraged to strengthen their discussion on the turbulent flux distributions, which may provide guidance for further effort on turbulence parameterizations under hurricane conditions. There are a few aspects worth attention. The authors are not asked to add a whole new section; one paragraph or two would be sufficient.

i) It would be interesting to apply the flux decomposition method to analyze the results from other experiments and compare the large-eddy-induced fluxes under the scenarios in which the single-mode roll structure is dominant or not.

ii) The vertical distributions of the roll-induced radial and tangential momentum fluxes presented in this study seem largely consistent with Gao and Ginis (2016), which investigated the correlation between the roll-induced momentum fluxes and the mean wind shear based on 2-D model results. It would be of interest to apply the same method and see if the 3-D LES results qualitatively agree with Gao and Ginis (2016).

Minor comments
Page 2 line 33: Suggest changing "Others neglect the effects by assuming  $\dots$ " to "Others neglect the horizontal advection effect on the HBL wind profiles by assuming  $\dots$ "

Page 3 line 4: Change "Morrison and Bussinger (2005)" to "Morrison et al. (2005)"

Page 4 line 7: The Charnock relationship gives a monotonic increase in the drag coefficient for increasing surface wind speeds, which was found not valid under high wind condition (surface wind greater than  $\sim$  30m/s). Did the authors put any constraint on the surface roughness length (drag coefficient) under high wind in this study?

Page 6 line 5: Foster (2005) used a linearized dynamical model to obtain the wind profiles and did not use observed wind.

Page 6 line 24: The fact that experiment H3 has the highest mixed layer is likely partially due to the strongest nonlocal mixing effect of rolls/large eddies, which have largest vertical extent in this experiment, not only due to the strongest turbulence intensity.

Page 7 lines 30-32 and Page 10 line 1: It is not clear why the authors say H3 has the most vigorous rolls. Is this based on the maximum w' or the domain-integrated kinetic energy? The turbulence statistics shown in Fig. 7 suggest that the maximum w' in L3 (which has strongest radial shear) maybe larger than H3.

Page 8 line 17: This sentence needs to be revised since it is somehow counterintuitive by saying "downward transport driven by vigorous upward motion".

Page 15 line 10: It is not clear how the tangential momentum flux w'v' affect the turbulence intensity.

Page 17 line 23: While it is true that rotation terms have no significant direct impact on rolls generated by the shear instability, Foster (2005) suggested that at small radii (insider of the radius of maximum wind) another type of instability associated with the rotation terms (the parallel instability) may be the dominant mechanism for roll formation. **ACPD**
Figures 3 and 4: At what time are these snapshots selected?

Figure 4: There is no black line in (c). Also, the caption for (d) should be w'u'.

---

## Referee Comment (RC2) · R. Foster (Referee) · 2 Dec 2016

Title: Impact of vertical wind shear on roll structure in idealized hurricane boundary layers

Authors: Shouping Wang, Qingfang Jiang

Recommend: Minor revisions

Synopsis: This paper describes a numerical model that can be used to investigate how the hurricane boundary layer mean flow affects the resulting organized roll vortices. Observations suggest that rolls are a fundamental aspect of the hurricane boundary layer. Yet, no numerical model PBL parameterizations include their effects. In large

part this is because quantitative information on how they affect the fluxes is lacking in observations. So, there have been attempts to capture their effects in LES-like numerical models. Given a simulation of rolls, their effects can be examined in isolation. The key question is whether or not the simulation properly represents the hurricane boundary layer.

Discussion: The method in this paper uses a relaxation methodology to impose a mean flow. It is clear that rolls are generated by a combination of shear and convective instabilities (dominated by the shear instability). The shape of the mean inflow profile (radial flow) controls the shear instability. However, the basic mean flow profiles are established by both the rolls and non-roll fluxes, which leaves a chicken-or-the-egg theoretical problem. Theory shows that the mean flow modifications due to rolls (roll-flux divergence effects) is somewhat subtle, so it makes sense to start with a reasonable non-roll flux (i.e. down-gradient, local fluxes), idealized PBL and examine the finite perturbations. This is the approach used in theoretical studies and it is essentially what has been done in this paper.

However, there is no clear way to select the down-gradient model and, as has been demonstrated elsewhere, the resulting mean flow profiles can vary wildly, as will the associated rolls. The value of this paper is that the relaxation methodology "generalizes" the LES technique toward the freedom allowed in theoretical studies to explore the parameter space associated with the selection of mean flow profiles. However, care must be taken to ensure that the target mean flow profiles are in fact realizable. There must be a consistent effective eddy-viscosity profiles that would produce the target mean flow profiles. This latent eddy viscosity associated with the target mean flow profile and the non-roll fluxes should be part of this paper.

Interesting questions arise. In some sense, current remote sensing capabilities are making it much easier to measure basic roll characteristics and mean surface winds than many other aspects of the turbulent hurricane boundary, especially turbulent fluxes (much less being able to separate the fluxes into roll and non-roll contributions).

Can tools such as the model presented here (and theoretical models) combined with observations be used to find constraints on the various local "gradient-flux" methodologies used in numerical models?

How can the non-local roll fluxes be parameterized? The mass flux-like method proposed by Zhu has a fundamental limitation. Convective boundary layers are comparatively much simpler than mixed shear/convective boundary layers. The former is highly skewed and the nonlocal fluxes are largely vertical over the locally warmer perturbations. The roll PBL is much less skewed, the roll characteristics depend on the entire mean flow profile, and, the updrafts are not vertical and co-located with the warmer temperature perturbations. (Zhu's lateral momentum entrainment parameter is an order of magnitude larger than that used for heat in mass flux models; this result has been replicated in theoretical models.) It might be simpler to attempt to modify the simple Mellor-Yamada-based parameterizations to include roll effects. However, the basic closure assumptions used in these models all assume near isotropy in the turbulence, so there may be fundamental inconsistencies.

In any case, the model that was developed for this paper stakes out an interesting middle path for exploring hurricane boundary layer rolls. In combination with the other tools and remote sensing capabilities that have been developed recently, we may be on the cusp of developing hurricane boundary parameterizations that correctly capture both the local and nonlocal contributions to the turbulent fluxes. As shown in this paper and elsewhere, the nonlocal fluxes are not a small contribution to the total. And they are inconsistent with standard down-gradient parameterization.

Specific suggestions:

1) Review the paper for English grammar. It is generally quite well written, but a few minor errors are present.

2) On page 10, line15-16 (and other places): I think you mean the "vertical shear of the radial wind".

3) Page 11 line 3: Might be clearer to use "r, t" subscripts instead of "x, y".

4) I think you need to provide effective eddy viscosity profiles for the choices of the target mean flow profiles, especially the "mixed" ones. As a further benefit, this would allow direct comparisons with theoretical roll models.

5) The roll effects are studied in detail and it took me a long time to puzzle through all of the figures. That isn't a criticism, there is a lot of information to digest. I wonder if some sort of "cartoon" drawing could help with the visualization and putting the results in context?

---

## Author Comment (AC1) · 26 Jan 2017

The comment was uploaded in the form of a supplement:
http://www.atmos-chem-phys-discuss.net/acp-2016-827/acp-2016-827-AC1-supplement.pdf

---

## Author Comment (AC2) · 26 Jan 2017

Response to Dr. Foster's comments

The authors appreciate Dr. Foster's insightful review and many helpful comments. A point-by-point response is provided below.

**Reviewer's comments:**

*Synopsis: This paper describes a numerical model that can be used to investigate how the hurricane boundary layer mean flow affects the resulting organized roll vortices. Observations suggest that rolls are a fundamental aspect of the hurricane boundary layer. Yet, no numerical model PBL parameterizations include their effects. In large part this is because quantitative information on how they affect the fluxes is lacking in observations. So, there have been attempts to capture their effects in LES-like numerical models. Given a simulation of rolls, their effects can be examined in isolation. The key question is whether or not the simulation properly represents the hurricane boundary layer.*

*Discussion: The method in this paper uses a relaxation methodology to impose a mean flow. It is clear that rolls are generated by a combination of shear and convective instabilities (dominated by the shear instability). The shape of the mean inflow pro- file (radial flow) controls the shear instability. However, the basic mean flow profiles are established by both the rolls and non-roll fluxes, which leaves a chicken-or-the-egg theoretical problem. Theory shows that the mean flow modifications due to rolls (roll-flux divergence effects) is somewhat subtle, so it makes sense to start with a reasonable non-roll flux (i.e. down-gradient, local fluxes), idealized PBL and examine the finite perturbations. This is the approach used in theoretical studies and it is essentially what has been done in this paper.*

*However, there is no clear way to select the down-gradient model and, as has been demonstrated elsewhere, the resulting mean flow profiles can vary wildly, as will the associated rolls. The value of this paper is that the relaxation methodology "generalizes" the LES technique toward the freedom allowed in theoretical studies to explore the parameter space associated with the selection of mean flow profiles. However, care must be taken to ensure that the target mean flow profiles are in fact realizable. There must be a consistent effective eddy-viscosity profiles that would produce the target mean flow profiles. This latent eddy viscosity associated with the target mean flow profile and the non-roll fluxes should be part of this paper.*

*Interesting questions arise. In some sense, current remote sensing capabilities are making it much easier to measure basic roll characteristics and mean surface winds than many other aspects of the turbulent hurricane boundary, especially turbulent fluxes (much less being able to separate the fluxes into roll and non-roll contributions). Can tools such as the model presented here (and theoretical models) combined with observations be used to find constraints on the various local "gradient-flux" methodologies used in numerical models?*

*How can the non-local roll fluxes be parameterized? The mass flux-like method pro- posed by Zhu has a fundamental limitation. Convective boundary layers are comparatively much simpler than mixed shear/convective boundary layers. The former is highly skewed and the nonlocal fluxes are largely vertical over the locally warmer perturbations. The roll PBL is much less skewed, the roll characteristics depend on the entire mean flow profile, and, the*

*updrafts are not vertical and co-located with the warmer temperature perturbations. (Zhu's lateral momentum entrainment parameter is an or- der of magnitude larger than that used for heat in mass flux models; this result has been replicated in theoretical models.) It might be simpler to attempt to modify the simple Mellor-Yamada-based parameterizations to include roll effects. However, the basic closure assumptions used in these models all assume near isotropy in the turbulence, so there may be fundamental inconsistencies.*

*In any case, the model that was developed for this paper stakes out an interesting middle path for exploring hurricane boundary layer rolls. In combination with the other tools and remote sensing capabilities that have been developed recently, we may be on the cusp of developing hurricane boundary parameterizations that correctly capture both the local and nonlocal contributions to the turbulent fluxes. As shown in this paper and elsewhere, the nonlocal fluxes are not a small contribution to the total. And they are inconsistent with standard down-gradient parameterization.*

**Response:**

We agree with the reviewer on many points in the discussion comments, particularly on the parameterization of the momentum flux from both the roll and non-roll contributions. In our view, there are at least two scenarios in which the K closure would fail. The first is related to the non-local flux problem for which the flux is no longer locally downgradient as described by the closure theory. This scenario usually occurs when the mean wind profile has a maximum or minimum across which the vertical gradient changes sign while the momentum flux maintains the same sign. For example, the super-gradient wind in HBL may lead to this condition (Gao and Ginis 2016; referred to as GG16 hereafter). The second occurs when there is a large difference between the transfer coefficients for the radial and tangential winds, implying a single momentum flux transfer coefficient does not work for both directions simultaneously. Both scenarios have been examined by multiple authors (e.g. GG16; Green and Zhang 2015). Our simulation (H3) has indications of both scenarios, although they seem to be less robust than in other studies.

One of the advantages of this mean nudging approach is to provide more flexibility in choosing various idealized and realistic mean wind profiles for the HBL. This work has demonstrated some success of this approach. On the other hand, because of the strong nudging, the effects of the rolls on the mean wind profile are difficult to isolate. This drawback is briefly touched upon in the last paragraph of the summary section. One of the issues the reviewer raised is whether the target wind (and therefore the LES mean wind) is realizable. This is a good and valid point. While our target wind profiles are not derived directly from any balanced HBL model solutions, they are carefully chosen based on the normalized profiles of Foster (2005) and the observations of Morrison et al (2005). Our overall comment is that the target winds have the essential HBL features so that the simulated rolls are consistent with observations and other theoretical studies in many aspects. The choices of the target winds are justified in this idealized HBL study even though they may not necessarily be observable in the real world or derivable from some basic balance dynamic systems. The detailed response is as follows.

The relaxation is used to nudge the LES mean wind toward the target profiles so that the result-ant equilibrium mean wind profiles have special characteristics for effective simulation comparisons.  For example, the three simulations in either group H or L have a similar inflection point height but different wind shears.  In general, group L has stronger wind shear and lower inflection point levels than those from group H.  In addition, all the simulations have about the same surface wind speed.  This is designed to investigate the impact of both the wind shear strength and the inflection point level associated with the shear on roll formation.  Because the target winds can never be reached in simulations, it is not possible to derive the eddy viscosity associated with the target winds from the LES simulations.  Although our target winds may be different from the basic-state mean wind profiles derived from the HBL momentum balance equations (e.g. Foster, 2005), they carry some essential features that are similar to the model derived and observed wind profiles, namely an inflection point in the radial wind profile, the super-gradient wind in HBL, and the gradient wind balance above the HBL.  Therefore, we believe that the target wind profiles used in our study are relevant to real HBLs and as a result, the simulated rolls have broad similarities with those from observations and theoretical studies.

To clarify this issue, we have expanded the description and discussion about how the target wind profiles are formulated for different simulations in the text.  Although we are not able to show the eddy-viscosity profile for the target wind, we present the eddy-viscosity profiles from three spectral groups for the LES mean winds of the simulation H3 in a new subsection 4.4.  New discussions are also included to reflect the above response.

**Modified text**
"The target wind profiles are formulated based on the normalized typical hurricane wind profiles obtained from a dynamical model of Foster (2005) and from the observations by Morrison et al. (2005).  The relaxation is used to nudge the LES mean wind toward the target profiles so that the resultant equilibrium mean wind profiles have special characteristics for effective simulation comparisons.  We have experimented with dozens of LES simulations using a variety of the wind profiles. The two groups of the target wind profiles (i.e., H and L groups, see Fig. 1) are chosen from these additional trial simulations and they exhibit systematic variations in shear strength and infection point levels to serve our objectives.  The target radial wind $U_T$ of H2 and tangential wind $V_T$ of group H generally follow those of Fig. 2 of Foster (2005) except for the HBL height. In addition, the super gradient wind shape is also included in $V_T$ in accordance to Fig 3a of Morrison et al. (2005).  The $U_T$ profile of H2 is multiplied by 0.5 and 1.5 to provide $U_T$ for H1 and H3 with the different shear strength but similar IPL, respectively.  The target radial wind $U_T$ of L2 is obtained by vertically suppressing $U_T$ of H2 and increasing the near-surface value to 13 m s$^{-1}$.  Then, $U_T$ of L2 is multiplied by 0.5 and 1.5 to give $U_T$ of L1 and L3, respectively. The target tangential wind profile $V_T$ of group L is obtained by lowering the HBL height for $V_T$ of group H." (Page 6, 1-12)

"While there is some quantitative difference between the target wind profiles defined above and the ones derived from the basic HBL balance equations such as those of Foster (2005), they carry some essential features that are similar to the model-derived or observed wind profiles such as an inflection point in the radial wind, the super-gradient wind in HBL, and the gradient wind balance above the HBL.  Given our objective of investigating the impact of the wind shear (including both the shear strength and the inflection point level) on the roll structure, we believe that our choices of the target winds are justified in the sense that they retain the basic HBL mean wind

features and provide a simple way to make a meaningful comparative study." (Page 6, line 23-29)

Please see the included subsection 4.4 in the response to the specific comment.

Specific suggestions:

*1) Review the paper for English grammar. It is quite well written, but a few minor errors are present.*

We have reviewed the paper very carefully.

*2) On page 10, line 15-16 (and other places): I think you mean the vertical shear of the radial wind".*

We have changed the wording according the reviewer.

" …increasing the vertical shear of the radial wind results in…"

*3) Page 11 line 3: Might be clearer to use "r, t" subscripts instead of "x, y".*

Because "*x*" and "*y*" are defined as the radial and tangential direction on page 3 (line 30) and page 4 (line 1-2) when the coordinate was introduced. For consistency, we still use "*x, y*" here. But the following sentence is added:

"(Note that the subscript "*x*" and "*y*" represents the radial and tangential direction, respectively, as defined in subsection 2.1.)"

*4) I think you need to provide effective eddy viscosity profiles for the choices of the target mean flow profiles, especially the "mixed" ones. As a further benefit, this would allow direct comparisons with theoretical roll models.*

As explained for the discussion comment, the target mean wind in the relaxation is only used to regulate the LES predicted mean wind to achieve the desired mean wind profiles. The target wind profiles are never reached in the simulations. Therefore, there is no effective eddy-viscosity for the target wind profiles from the simulations. We, however, present the eddy-viscosity profiles from three spectral groups for the LES mean winds of the simulation H3 in a new subsection 4.4. In our view, it is the LES mean wind (not the target wind) that directly generates the turbulence and roll circulations. Therefore, these eddy-viscosity and the mean wind profiles may be tested in the theoretical roll models. The entire new subsection is copied here.

**"4.4 Momentum transfer coefficients**

Momentum transfer coefficients, defined by the negative ratio of the momentum flux to the mean wind shear according to the $K$ theory, play a central role in the representation of HBL. It has

[revised manuscript text omitted]

***5)   The roll effects are studied in detail and it took me a long time to puzzle through all of the figures.  That isn't a criticism; there is a lot of information to digest.  I wonder if some sort of "carton" drawing could help with the visualization and putting the results in context.***

We understand the reviewer's frustration.  It would be certainly nice to summarize these results in a "carton" drawing.  We have made further effort to improve the text and figure captions. We have come up with several versions of "carton" to help summarizing the dynamics and roll characteristics, as suggested by the reviewer. Unfortunately, none of them are artistic and revealing enough to our satisfaction.  It is probably, in part, because the work describes several aspects of the rolls such as the mean wind profiles, instantaneous roll patterns, roll spectral characteristics, etc.  It is really difficult to put together these various aspects in a cartoon.

---

## Author Comment (AC3) · 27 Jan 2017

[revised manuscript text omitted]
 value of 45.5 m s$^{-1}$ is used as it represents a middle-to-high speed range of the gradient wind for a hurricane environment (e.g. Willoughby, 1990).

The target wind profiles are formulated based on the normalized typical hurricane wind profiles obtained from a dynamical model of Foster (2005) and from the observations by Morrison et al. (2005). The relaxation is used to nudge the LES mean wind toward the target profiles so that the resultant equilibrium mean wind profiles have special characteristics for effective simulation comparisons. We have experimented with dozens of LES simulations using a variety of the wind profiles. The two groups of the target wind profiles (i.e., H and L groups, see Fig. 1) are chosen from these additional trial simulations and they exhibit systematic variations in shear strength and infection point levels to serve our objectives. The target radial wind $U_T$ of H2 and tangential wind $V_T$ of group H generally follow those of Fig. 2 of Foster (2005) except for the HBL height. In addition, the super gradient wind shape is also included in $V_T$ in accordance to Fig 3a of Morrison et al. (2005). The $U_T$ profile of H2 is multiplied by 0.5 and 1.5 to provide $U_T$ for H1 and H3 with the different shear strength but similar IPL, respectively. The target radial wind $U_T$ of L2 is obtained by vertically suppressing $U_T$ of H2 and increasing the near-surface value to 13 m s$^{-1}$. Then, $U_T$ of L2 is multiplied by 0.5 and 1.5 to give $U_T$ of L1 and L3, respectively. The target tangential wind profile $V_T$ of group L is obtained by lowering the HBL height for $V_T$ of group H.

In summary, group L simulations are forced with the target radial wind profiles ($U_T$) that have three shear strengths with IPLs approximately at 200 m (Fig. 1a). Similarly, group H simulations also have three shear strengths with IPLs between 400 m–500 m in the radial wind. The target tangential wind profile ($V_T$) is specified in Fig. 1b. The $V_T$ profile with the shear occurring below 700 m (dash-dotted) is used for group L simulations, the other (solid) for group H. This paper is focused on the radial wind shear because of its direct link to the inflection instability (GG14). Therefore, only one target tangential wind is prescribed for each simulation group, which has three target radial wind profiles as discussed above. It is recognized that changes in the radial wind inevitably affect the tangential wind. The sensitivity of the LES results to the tangential winds is also explored. The simulations and relevant parameters are listed in Table 1.

While there is some quantitative difference between the target wind profiles defined above and the ones derived from the basic HBL balance equations such as those of Foster (2005), they carry some essential features that are similar to the model-derived or observed wind profiles such as an inflection point in the radial wind, the super-gradient wind in HBL, and the gradient wind balance above the HBL. Given our objective of investigating the impact of the wind shear (including both the shear strength and the inflection point level) on the roll structure, we believe that our choices of the target winds are justified in the sense that they retain the basic HBL mean wind features and provide a simple way to make a meaningful comparative study.

**3 Overall turbulence structure**

This section is centred on a comparison of turbulence fields and statistics between group L and H simulations (see Table 1). A special attention is given to the roll structure manifested by the coherent and organized turbulent flow. All the profiles presented here are obtained from ensemble averaging, which is applied over the entire horizontal domain and between 8 and 10 h with a sample interval of 30 s.  A time series of an average variable is constructed by taking the horizontal mean every minute.

**3.1 Time evolution and mean state**

To gain a general impression of the HBL development and differences among the simulations, the time series of the HBL heights ($z_i$) and mean profiles are examined.  As shown in Fig. 2, $z_i$ increases rapidly with time for most simulations during the first 5 hours, after which the growth rate slows down considerably, implying a quasi-equilibrium state being reached.  H2 behaves slightly differently in that $z_i$ becomes slowly varying only after 8 h.  There is a clear tendency of a stronger radial wind shear resulting in higher $z_i$ for each group ($L$ or $H$) of simulations.  The lowest $z_i$ is obtained by L1 and H1, whose turbulence intensities are low because of the weak radial wind shear for both cases. It is worth noting that H3 predicts the highest $z_i$ among all the simulations, suggesting that it produces the strongest turbulence intensity even though it does not have the strongest radial shear (Table 1).  It will be shown in following sections that H3 results in the most vigorous roll structure, which is likely to contribute to the highest $z_i$ by 
[revised manuscript text omitted]

---

## Author Comment (AC4) · 27 Jan 2017

Dear Editor,

The "marker-up" version of the manuscript is uploaded. Please note that I have not used the track change in the manuscript. Instead, I simply highlighted the changes with yellow color (I may have missed some minor ones). Hope this is acceptable.

Thanks, Shouping